# Self-Explainable Molecular Property Prediction via Multi-view Hypergraph Learning

## Abstract

Explainable molecular property prediction plays a critical role in drug discovery and materials science. Existing graph neural network (GNN)-based approaches usually rely on pairwise atomic interactions for molecular modeling and interpretation. However, such atom-centered modeling often neglects the cooperative effects of atomic groups and lacks the guidance of chemical rules, thereby limiting both prediction accuracy and interpretability. To address these challenges, we propose a multi-view **Hyper**graph learning method for **S**elf-**E**xplainable **M**olecular property prediction (HyperSEM). Our method introduces a hyperedge-driven explanation paradigm, where atomic groups are explicitly modeled as hyperedges to capture high-order cooperative effects, and multi-view hypergraphs are constructed to jointly integrate chemical rules and data-driven signals. Furthermore, we design a molecular structure-informed hypergraph convolution to preserve both high-order atomic-group interactions and low-order structural features, and an information-bottleneck-guided self explanation to jointly generate predictions and explanations. Extensive experimental results show that HyperSEM outperforms existing state-of-the-art methods on seven benchmark datasets, demonstrating dual advantages in prediction accuracy and interpretability.

## 1 Introduction

Molecular property prediction plays a crucial role in biopharmaceutical development and novel material design, enabling key tasks such as drug screening, toxicity assessment, and material synthesis Wieder et al. (2020); Vamathevan et al. (2019). By modeling molecules as graphs with atoms as nodes and bonds as edges, graph neural networks (GNNs) have achieved state-of-the-art performance in molecular property prediction by jointly capturing local atomic environments and global structural topology Li et al. (2022). Despite their promising performance, most GNN-based approaches lack interpretability, i.e., they can only predict final outcomes but fail to reveal which underlying chemical substructures contribute to these results. The absence of such interpretability limits their deployment in several real-world scenarios such as drug discovery and toxicity assessment, where transparent and trustworthy predictions are critical Jiménez-Luna et al. (2020).

A potential solution to fill the above gap is explainable GNN methods, which aim to provide insights into model decisions of GNNs by identifying the most influential substructures that contribute to the predictions Ying et al. (2019); Yuan et al. (2021). Through parameterizing the nodes and/or edges with learnable scores, these methods estimate their importance via gradient-based optimization of the scores, and hence highlight the most significant nodes/edges with the highest scores as key explanations for the predictions Luo et al. (2020). In order to ensure the consistency between predictions and explanations, the existing explainable GNN methods usually rely on a data-driven learning mechanism to optimize the importance score of each node or edge, where the scores are adjusted to align with the predictive behavior of models. For instance, the nodes and edges that can maximize the predicted probability are prone to being identified as important contributors Vu & Thai (2020).

Despite their effectiveness, these methods are designed for general GNN models rather than tailored for molecular property prediction, and thus fall short in offering chemically meaningful explanations. Specifically, a critical factor in chemistry is that molecular properties are often determined by cooperative effects among multiple atoms Hansch & Fujita (1964); Veber et al. (2002) and by the influence of functional groups Lee et al. (2025); Kalgutkar (2019), which cannot be captured by

atom- or bond-level modeling alone. Such a mismatch gives rise to two major limitations of existing GNN explanation methods when handling molecular property prediction problems: *neglect of atom group* and *ignorance of chemical rule*.

*Limitation 1 -* **neglect of atom groups**. To achieve fine-grained explanations, most existing explanation approaches treat nodes or edges as their basic explanation unit, which may overlook higher-order cooperative interactions among multiple atoms and thus fail to capture the influence of atom groups on molecular properties. Concretely, the explanatory unit can solely characterize the contribution of an atom or a bond in a molecular graph, which neglects the combination of multi-atom

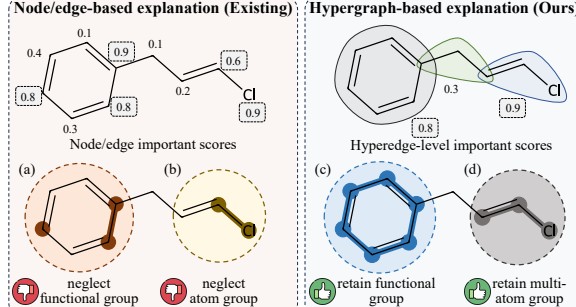

Figure 1: Comparison of explanation paradigms.

effects, such as the electron-withdrawing interaction between the nitrogen atom in a nitro group and its bonded carbon atoms in nitrobenzene. Without a comprehensive consideration of all atomic components in an atom group, the explanation approaches may only identify a fraction of relevant atoms/bonds as the explanation, and hence provide incomplete explanations. As illustrated in Figure 1(b), due to their lack of multi-atom modeling capabilities, conventional methods may misinterpret the cooperative effect of a C=C double bond and a chlorine atom as isolated interactions between the interaction between Cl and C, or the isolated effect of the C=C double bond, overlooking the combined electronic effect of chlorine's strong electron-withdrawing property and the double bond. This misinterpretation not only undermines the interpretability of the model, leading to inaccurate molecular property predictions.

*Limitation 2 -* **ignorance of chemical rules**. Without the guidance of domain knowledge, current explainable methods are usually data-driven, where the contributions of each graph component are inferred purely from their neural correlations with the final prediction. However, purely data-driven methods cannot ensure that the generated explanations align with the chemical structure, resulting in interpretations that contradict chemical prior knowledge. Moreover, without effective modeling of chemical principles, they fail to accurately capture the cooperative interactions between atom groups (i.e., functional group), thus further impacting the accurate prediction of molecular properties. For example, traditional methods may misinterpret an amide group as separate amine groups (see Figure 1(a)), which overlooks their synergistic effects. Such misinterpretation can compromise the chemical validity of the explanations and reduce their utility in molecular property identification.

To address the above limitations, in this paper, we propose a novel multi-view **Hyper**graph learning method for **S**elf-**E**xplainable **M**olecular property prediction (HYPERSEM). Our key innovation is to model each molecule as a hypergraph, with well-defined hyperedges providing chemically faithful and structurally coherent explanations (See Figure 1(c,d)). Specifically, to address *Limitation 1*, we explicitly model different atom groups into hyperedges to capture high-order cooperative interactions among multiple atoms. This design enables the model to focus on holistic substructures that are often decisive for molecular properties. Moreover, to handle *Limitation 2*, we design a multi-view hypergraph construction strategy that jointly integrates chemical rules and data-driven signals into the molecular property prediction and explanation process. To be concrete, we explicitly incorporate chemical rules via the BRICS algorithm to generate hyperedges, while also preserving local structure awareness through local-based hyperedges and maintaining data-driven adaptability through dynamically constructed hyperedges. To alleviate the degradation of topological resolution commonly introduced by high-order hypergraph connections, we propose a molecular structure-informed hypergraph convolution model that retains both local and global structural information during message passing. In addition, an interpretation module is integrated at the hyperedge level, allowing the model to score the importance of atom groups and generate interpretable substructure-level explanations aligned with chemical intuition. To sum up, our contribution can be summarized as follows:

- **New Explanation Paradigm.** Going beyond the conventional node/edge-based explanation paradigm, we propose a hyperedge-based paradigm that captures higher-order cooperative interactions among multiple atoms and provides chemically meaningful explanations.

- **Novel Self-Explainable Method.** Following the new paradigm, we design HYPERSEM, a well-crafted method for self-explainable molecular property prediction. HYPERSEM incorporates a multi-view hypergraph construction to integrate chemical rules and data-driven signals, with a molecular structure-informed hypergraph convolution for higher-order structural modeling and an information bottleneck module for concise explanations.
- **Extensive Experiments.** We conduct ample experiments on multiple benchmark datasets to validate the effectiveness and interpretability of HYPERSEM. The experimental results demonstrate its superior predictive performance and more chemically faithful explanations compared to state-of-the-art baselines.

## 2 RELATED WORK

**Molecular Property Prediction** (MPP) is a fundamental task in drug discovery and materials design. Early methods, including fingerprint-based methods Rogers & Hahn (2010) and sequence-based methods Goh et al. (2017); Chithrananda et al. (2020); Wang et al. (2019), usually ignore explicit structural topology and fail to capture complex molecular interactions. Recently, graph neural networks (GNNs) have therefore become the dominant paradigm for MPP. Pure message passing methods like Jo et al. (2022); Gasteiger et al. (2020) capture atom- and bond-level dependencies directly from molecular graphs. More advanced designs incorporate chemical priors or auxiliary information; for instance, methods like Cai et al. (2022); Zhang et al. (2024) augment graph representations with molecular fingerprints. Beyond pairwise graph modeling, hypergraph-based methods have been explored to capture higher-order substructures Liu et al. (2025); Cui et al. (2023). Despite their effectiveness, these methods can only provide predictions for molecular properties but fail to provide explanations for their predictions, which hinders their scientific applicability.

**Explainable Graph Neural Networks** have become a key research area to enhance the reliability of GNN models. Existing methods can be broadly categorized into post-hoc and built-in approaches. In the post-hoc category, GNNExplainer Ying et al. (2019) is a representative approach that identifies crucial subgraphs and node features by optimizing a mutual information objective. Follow-up studies Luo et al. (2020); Vu & Thai (2020); Yuan et al. (2021) follow a similar paradigm but use different techniques to refine the optimization process and achieve reliable explanations. There are also generative methods for graph explanation. Gem Lin et al. (2021) and MAGE Yu & Gao (2024) generate class-relevant subgraphs for explanation, though they typically require substantially higher computational cost. More recently, subgraph matching Wu et al. (2023) and pruning Wang et al. (2021) are also proposed for explanation. For built-in approaches (a.k.a. self-explainable methods) Rao et al. (2024); Zhang et al. (2022); Lucic et al. (2022), the explanation modules are built into the prediction models and can generate interpretations jointly with predictions in an end-to-end manner. Despite their progress, most existing methods still operate at the level of individual nodes or edges. Even those extending to subgraph or prototype explanations often fail to capture chemically cooperative effects among multiple atoms and functional groups.

## 3 METHODOLOGY

In this paper, we propose HYPERSEM, a novel self-explainable molecular property prediction approach based on multi-view hypergraph learning. The theme of HYPERSEM is to capture higher-order cooperative interactions among atoms and functional groups through multi-view hypergraph representations, thereby enabling both predictive accuracy and providing hyperedge-level interpretability. As shown in Figure 2(a), HYPERSEM consists of three components: (1) a *multi-view hypergraph construction* module that builds hyperedges through static and dynamic strategies to model atom groups and incorporate chemical rules; (2) a *molecular structure-informed hypergraph convolution network* that learns hypergraph representations for molecular samples; and (3) an *information bottleneck-guided self-explanation* module that identifies informative hyperedges to enhance interpretability and generalization. ALL components are elaborated in the following subsections. A detailed pseudo code is given in Appendix A and an efficiency study is provided in Appendix B.

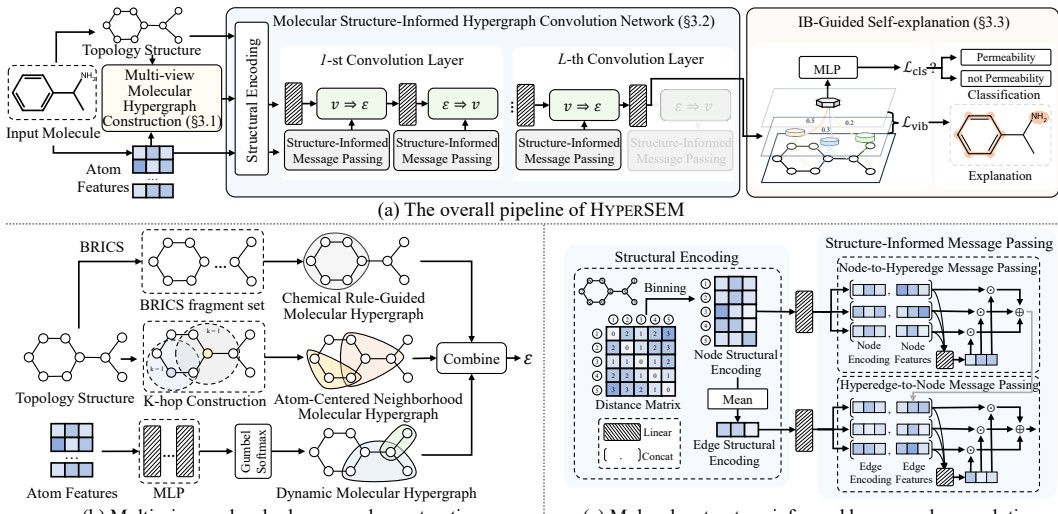

Figure 2: The overall pipeline and key components of HYPERSEM.

## 3.1 MULTI-VIEW MOLECULAR HYPERGRAPH CONSTRUCTION

Due to the intrinsic relational nature of molecular structure, it is natural to model a molecule as a graph $\mathcal{G} = (\mathcal{V}, \mathcal{E}_{\text{bond}})$, where $\mathcal{V} = \{v_1, v_2, \ldots, v_N\}$ is the set of $N$ atoms, and $\mathcal{E}_{\text{bond}}$ denotes the set of chemical bonds. Despite the success of existing works adopting this paradigm Jo et al. (2022); Gasteiger et al. (2020); Zhang et al. (2024), they often ignore higher-order cooperative interactions among multiple atoms and functional groups, which leads to limited interpretability and suboptimal predictive performance. To address this limitation, hypergraph-based modeling provides a more suitable solution, since hyperedges can explicitly model atom groups and capture complex many-to-many relationships beyond simple pairwise bonds. The hyperedges can not only provide chemically meaningful interpretations but also contribute to more accurate property predictions.

While hyperedges play a central role in HYPERSEM, how to construct these chemically meaningful hyperedges remains a fundamental challenge. On the one hand, the hyperedges should obey chemical rules that reflect functional groups and substructural motifs and cover local multi-atom interactions; on the other hand, it is also crucial to learn task-relevant hyperedges dynamically from data. To jointly leverage these complementary views, we design a multi-view hypergraph $\mathcal{H} = \{\mathcal{V}, \mathcal{E}\}$ for each molecule whose hyperedge set is composed of three parts: $\mathcal{E} = \mathcal{E}_{\text{rule}} \cup \mathcal{E}_{\text{khop}} \cup \mathcal{E}_{\text{dyn}}$.

**Chemical Rule-Guided Molecular Hypergraph Construction.** To explicitly encode chemical prior knowledge into the model, we require hyperedges that directly correspond to chemically meaningful substructures. To this end, we adopt the **BRICS** algorithm Degen et al. (2008) to perform chemically reasonable fragmentation of a molecule. BRICS defines a series of robust bond-breaking rules grounded in retrosynthetic chemistry. By applying these rules on the molecular graph $\mathcal{G}$, we decompose $\mathcal{G}$ into a set of fragments $f_1, f_2, \ldots, f_M$. For each fragment $f_i$, we directly define a hyperedge $e_i \in \mathcal{E}_{\text{rule}}$:

$$\mathcal{E}_{\text{rule}} = \{e_i \mid e_i = \text{GetAtoms}(f_i), f_i \in \text{BRICS}(\mathcal{G})\}, \tag{1}$$

where $\text{BRICS}(\mathcal{G})$ denotes the set of fragments obtained by applying BRICS rules to $\mathcal{G}$, and $\text{GetAtoms}(f_i)$ returns the set of atoms contained in fragment $f_i$. This construction ensures that hyperedges are not only structurally connected but also exhibit strong chemical cohesion in terms of functional semantics.

**Atom-Centered Neighborhood Molecular Hypergraph Construction.** Considering that atoms and their local chemical environments play a decisive role in molecular properties, we construct hyperedges by capturing the local neighborhood centered on each atom. Specifically, we employ an atom-wise k-hop neighborhood sampling strategy: for each atom $v_i \in \mathcal{V}$ in the molecular graph $\mathcal{G}$, its $k$-hop neighborhood $\mathcal{N}_k(v_i)$ is defined as a hyperedge. Here, the $k$-hop neighborhood refers to the set of atoms (including $v_i$ itself) that can be reached from $v_i$ within at most $k$ chemical bonds.

Formally, the collection of such hyperedges is defined as:

$$\mathcal{E}_{\text{khop}} = \{e_i \mid e_i = \mathcal{N}_k(v_i), \forall v_i \in \mathcal{V}\}, \tag{2}$$

where $\mathcal{N}_k(v_i) = \{v_j \in \mathcal{V} \mid \text{dist}(v_i, v_j) \leq k\}$. By tuning the hyperparameter $k$, the model is able to capture local atomic interactions at different scales.

**Dynamic Molecular Hypergraph Construction.** While the above two types of static hyperedges can capture basic chemical constraints and local multi-atom interactions, they are limited to local connectivity and fail to capture cross-group or long-range dependencies that are critical for molecular property prediction. For instance, given molecules such as paracetamol, the joint resonance effect on the aromatic $\pi$-system cannot be represented. Moreover, the static hyperedges lack data-driven adaptability to capture diverse molecular contexts. To address this issue, we introduce dynamically learnable hyperedges that adaptively capture supplementary interactions. This is achieved through a light-weight learnable linear transformation layer $\mathbf{W}_{\text{dyn}} \in \mathbb{R}^{D \times N_{\text{dyn}}}$, where $N_{\text{dyn}}$ is the number of dynamic hyperedges. Specifically, for the feature vector $\mathbf{x}_i$ of each atom $v_i$, we compute its assignment score vector over the $N_{\text{dyn}}$ hyperedges as: $\mathbf{s}_i = \mathbf{x}_i \mathbf{W}_{\text{dyn}}$. Subsequently, we apply the Gumbel-Softmax to generate a differentiable approximation of discrete hyperedge membership:

$$h_{i,e} = \frac{\exp\left((\mathbf{s}_{i,e} + g_e)/\tau\right)}{\sum_{e'=1}^{N_{\text{dyn}}} \exp\left((\mathbf{s}_{i,e'} + g_{e'})/\tau\right)}, \quad g_e \sim \text{Gumbel}(0, 1), \tag{3}$$

where $\tau$ is the temperature parameter, $g_e$ is Gumbel noise, and $h_{i,e}$ denotes the soft assignment of atom $v_i$ to the $e$-th hyperedge.

## 3.2 MOLECULAR STRUCTURE-INFORMED HYPERGRAPH CONVOLUTION NETWORK

After the construction of the multi-view molecular hypergraph, the following step is to learn high-quality representations for atoms and atom groups (i.e., hyperedges) through a hypergraph convolution network. In conventional hypergraph convolution networks, the bond connectivity of molecules is often overlooked in the process of representation learning of hypergraph convolution. As a result, the hyperedges fail to adequately preserve the original topological cues of the molecular graph, which in turn hampers accurate molecular property prediction. To address this limitation, we propose a molecular structure-informed hypergraph convolution network that incorporates structural encoding to enhance the model's awareness of relative positions and connectivity patterns among nodes. With the integration of structural encoding, we further design a two-stage message passing mechanism to jointly capture node- and hyperedge-level structural dependencies. A sketch map is given in Figure 2(c).

**Structural Encoding.** In order to preserve the original molecular structure and enrich spatial context, we introduce a structural encoding in our hypergraph convolution network. In HYPERSEM, we adopt a shortest-path distance-based encoding, as the relative position between atoms can be characterized by the shortest-path distance. Specifically, given $\mathbf{A} \in \mathbb{R}^{N \times N}$ as the adjacency matrix of a molecule graph $\mathcal{G}$ with bond connection, the $(i, j)$-th entry of its $m$-th power, $\mathbf{A}_{i,j}^m$, denotes the number of paths of length $m$ from node $v_i$ to $v_j$. If $\mathbf{A}_{i,j}^m > 0$, it indicates the existence of such a path. Accordingly, the shortest-path distance is defined as:

$$\text{dist}(v_i, v_j) = \min\{m \mid \mathbf{A}_{i,j}^m \neq 0\}. \tag{4}$$

To effectively compress and represent these topological relationships within a hyperedge, we discretize the shortest-path distances into bins according to path length. Given $M$ as the maximum cutoff distance, for a hyperedge $e$, the count of the $m$-th bin is defined as:

$$N_{i,m}(e) = \sum_{v_i \in e, v_j \in e} \mathbb{I}(\text{dist}(v_i, v_j) = m), \tag{5}$$

For the number of nodes whose shortest-path distance exceeds $M$, we further use $N_{-1}(e)$ to denote. Finally, we concatenate these counts to form the raw structural encoding vector of hyperedge $e$:

$$\mathbf{t}_{i,e} = [\, N_{i,-1}(e), N_{i,0}(e), N_{i,1}(e), \ldots, N_{i,M}(e)\,] \in \mathbb{R}^{M+1}. \tag{6}$$

To further strengthen the representation capability of HYPERSEM, we introduce learnable mappings at both the node and hyperedge levels to obtain the final structural encoding vector $\mathbf{z}_{i,e}$:

$$\mathbf{z}_{i,e} = \mathbf{W}_{nt}\mathbf{t}_{i,e} + \mathbf{b}_{nt}, \quad \mathbf{z}_e = \mathbf{W}_e\left(\frac{1}{|e|}\sum_{v_i \in e}\mathbf{t}_{i,e}\right) + \mathbf{b}_e. \tag{7}$$

With the structural encoding, our hypergraph convolution effectively preserves the molecular structural characteristics (i.e. bond connectivity) within each hyperedge. This bond information and the hyperedge-based multi-atom interactions form a complementary relationship, enabling more comprehensive molecular representations.

**Node-to-Hyperedge Message Passing.** Based on the above encoding, we propose a molecular structure-informed hypergraph convolution layer, which updates node representations through a two-stage message passing mechanism. This mechanism deeply integrates the previously defined structural encodings to enable precise perception of graph structures. Concretely, let $\mathbf{x}_i^l$ denote the feature of node $v_i$ at the $l$-th layer, the message passing process of this stage can be formulated as follows:

$$\alpha_{i,e} = \frac{\exp(\text{LeakyReLU}(\mathbf{a}_1^\top[\mathbf{W}_1\mathbf{x}_i^l\|\mathbf{W}_2\mathbf{z}_{i,e}]))}{\sum_{v_k \in e}\exp(\text{LeakyReLU}(\mathbf{a}_1^\top[\mathbf{W}_1\mathbf{x}_k^l\|\mathbf{W}_2\mathbf{z}_{i,e}]))}, \quad \mathbf{h}_e^l = \sigma(\sum_{v_i \in e}\alpha_{i,e}\mathbf{W}_{\text{node}}\mathbf{x}_i^l), \quad (8)$$

where $\mathbf{h}_e^l$ is the hidden representation of hyperedge $e$, $\alpha_{i,e}$ is attention weight, $\mathbf{W}_1$, $\mathbf{W}_2$, $\mathbf{W}_{\text{node}}$, and $\mathbf{a}_1$ are learnable parameters, and $\|$ denotes the concatenation operation.

The node-to-hyperedge message passing allows each hyperedge to adaptively capture the most informative node features while preserving topology structural cues, thereby enhancing the expressiveness of molecular representations.

**Hyperedge-to-Node Message Passing.** In this stage, the updated hidden representation of node $v_i$, denoted as $\mathbf{x}_i^{l+1}$, is aggregated from all hyperedges that contain it through hyperedge-to-node message passing. Similarly, we employ a molecular structure-informed gated attention mechanism to compute the contribution weight $\beta_{e,i}$ of hyperedge $e$ to node $v_i$. This weight is determined jointly by the aggregated feature of the hyperedge $\mathbf{h}_e^l$ and its global structural encoding $\mathbf{z}_e$:

$$\beta_{e,i} = \frac{\exp(\text{LeakyReLU}(\mathbf{a}_2^\top[\mathbf{W}_3\mathbf{h}_e^l\|\mathbf{W}_4\mathbf{z}_e]))}{\sum_{e_k:v_i \in e_k}\exp(\text{LeakyReLU}(\mathbf{a}_2^\top[\mathbf{W}_3\mathbf{h}_{e_k}^l\|\mathbf{W}_4\mathbf{z}_{e_k}]))}, \quad \mathbf{x}_i^{l+1} = \sigma(\sum_{e:v_i \in e}\beta_{e,i}\mathbf{W}_{\text{edge}}\mathbf{h}_e^l),$$

$$(9)$$

where $\mathbf{W}_3$, $\mathbf{W}_4$, $\mathbf{W}_{\text{edge}}$, and $\mathbf{a}_2$ are learnable parameters. The hyperedge-to-node stage complements the previous stage by closing the information loop, ensuring that local atom information is refined with hyperedge-level context.

## 3.3 Information Bottleneck-Guided Self-Explanation

To simultaneously make accurate predictions and provide faithful explanations, we develop an information bottleneck (IB)-guided self-explanation module, which integrates information-theoretic constraints with attention to achieve both accurate prediction and interpretable substructure discovery. Taking the high-quality hyperedge representations learned by the previous convolutional network as input, the self-explanation module can select the most informative hyperedges for prediction and interpretation.

**Hyperedge Scoring.** Given the set of hyperedge representations $\{\mathbf{h}_e^L, e \in \mathcal{E}\}$ obtained after $L$ layers of molecular structure-informed hypergraph convolution, we first compute an importance score for each hyperedge with respect to the target property via a learnable scoring function:

$$s_e = \sigma(\mathbf{w}_s^\top\mathbf{h}_e^L + b_s), \quad (10)$$

where $\mathbf{w}_s \in \mathbb{R}^D$ is the scoring vector, $b_s$ is the bias term, and $\sigma$ denotes the sigmoid activation function. $s_e$ indicates the contribution of hyperedge $e \in \mathcal{E}$ to the final prediction.

**Hyperedge Aggregation.** To obtain a molecule-level representation, we employ an attention-weighted selection mechanism that aggregates all hyperedge features into the global molecular representation $\mathbf{z}_{\text{mol}} \in \mathbb{R}^D$ based on the learned importance score $s_e$ for each hyperedge:

$$\mathbf{z}_{\text{mol}} = \sum_{e \in \mathcal{E}}\frac{s_e}{\sum_{e' \in \mathcal{E}}s_{e'}}\mathbf{h}_e^L. \quad (11)$$

The learned molecular representation $\mathbf{z}$mol can be further fed into a prediction head $f_\phi(\cdot)$ to produce the final molecular property predictions.

| Datasets (Tasks) | MoleculeNet | | | TUDataset | | | |
|---|---|---|---|---|---|---|---|
| Method | BBBP | BACE | HIV | AIDS | MUTAG | NCI1 | BZR |
| GIN | 94.3±3.4 | 73.5±3.4 | 91.6±3.4 | 75.8±3.4 | 92.1±3.4 | 77.2±0.6 | 73.3±3.4 |
| FP-GNN | 91.2±4.3 | 88.2±4.3 | 89.1±4.3 | 83.6±4.3 | 85.0±4.3 | 75.7±3.7 | 76.7±4.3 |
| FH-GNN | 91.6±2.0 | 89.1±2.0 | 82.5±2.0 | 89.4±2.0 | 93.0±2.0 | 77.2±2.1 | 77.4±2.0 |
| Graph-CBN | 75.8±2.1 | 83.6±2.3 | 73.8±2.3 | 86.9±2.3 | 94.0±2.3 | 74.6±2.5 | 74.2±2.3 |
| AttentionFP | 92.1±1.3 | 85.0±1.6 | 83.2±2.1 | 94.0±2.1 | 95.3±2.1 | 79.6±2.3 | 71.0±2.1 |
| PR-MPNN | 95.6±1.8 | 84.9±2.8 | 79.2±1.3 | 94.0±1.4 | 92.5±2.3 | 81.9±1.5 | 86.0±1.4 |
| MolCLR | 91.2±3.0 | 89.0±3.0 | 80.6±3.0 | 96.5±3.0 | 97.9±3.0 | 83.2±3.2 | 83.9±3.0 |
| HyperMol | 92.2±1.2 | 89.8±0.9 | 81.4±1.1 | 93.6±2.3 | 89.4±1.0 | 78.5±1.0 | 79.4±1.7 |
| MultiChem | 95.3±2.4 | **91.3±2.4** | 82.9±2.4 | 95.5±2.4 | **98.4±2.4** | 81.9±2.7 | 77.4±2.4 |
| HYPERSEM (Ours) | **97.2±2.0** | 90.4±0.7 | **84.6±1.6** | **99.7±0.2** | 98.4±2.1 | **86.1±0.6** | **87.1±2.0** |

Table 1: Comparison of molecular property prediction performance on MoleculeNet and TUDataset.

**IB-Guided Model Training.** To ensure that the model selects the most informative hyperedges, we introduce the information bottleneck regularization strategy. Specifically, we maximize the mutual information between the aggregated representation $\mathbf{z}_{\mathrm{mol}}$ and the target label $y$, while minimizing the redundancy between pre- and post-selection representations:

$$\mathcal{L}_{\mathrm{MI}} = -I(\mathbf{z}_{\mathrm{mol}}; y) + \lambda \cdot I(\{\mathbf{h}_e^L\}; \mathbf{z}_{\mathrm{mol}}), \tag{12}$$

where $I(\cdot; \cdot)$ denotes mutual information and $\lambda$ is a trade-off coefficient. For classification tasks, maximizing $I(\mathbf{z}_{\mathrm{mol}}; y)$ can be effectively achieved by minimizing cross-entropy loss. Since directly computing $I(\mathbf{h}_e^L; \mathbf{z}_{\mathrm{mol}})$ is intractable, we follow the Variational Information Bottleneck (VIB) framework and approximate it via a variational upper bound:

$$I(\{\mathbf{h}_e^{(L)}\}; \mathbf{z}_{\mathrm{mol}}) \leq \mathbb{E}_{p(\{\mathbf{h}_e^{(L)}\})}[\mathrm{KL}(p(\mathbf{z}_{\mathrm{mol}}|\{\mathbf{h}_e^L\})\|r(\mathbf{z}_{\mathrm{mol}}))] = \mathcal{L}_{\mathrm{vib}}, \tag{13}$$

with $r(\mathbf{z}_{\mathrm{mol}})$ denoting the prior distribution. The overall training objective thus combines the classification loss with the information-theoretic regularization:

$$\mathcal{L}_{\mathrm{total}} = \mathcal{L}_{\mathrm{cls}}(f_\phi(\mathbf{z}_{\mathrm{mol}}), y) + \beta \cdot \mathcal{L}_{\mathrm{vib}}, \tag{14}$$

where $\mathcal{L}_{\mathrm{cls}}$ is the cross-entropy loss and $\beta$ controls the regularization strength.

**Prediction and Interpretation.** During inference, the prediction head $f_\phi(\cdot)$ can output the predicted molecular properties given the learned representation. At the same time, the model can provides explanations through hyperedge importance scores $s_e$. Specifically, the atom groups (i.e. hyperedges) with higher scores can serve as as critical substructures that dominantly influence molecular properties, which offers chemically meaningful interpretability of the predictions.

## 4 EXPERIMENT

### 4.1 EXPERIMENTAL SETUP

**Datasets.** We evaluate our model on eight benchmark datasets: BBBP, BACE, and HIV from MoleculeNet Wu et al. (2018), and AIDS, MUTAG, BZR, NCI1, and BZR from TU Dataset Morris et al. (2020). Detailed dataset introduction is listed in Appendix D and experimental setup is in Appendix F.

**Baselines.** We compare our method with representative baselines. For classification, we consider GIN Xu et al. (2018), FP-GNN Cai et al. (2022), FH-GNN Liu et al. (2025), Deep-CBN Kianfar et al. (2025), AttentionFP Lei et al. (2022), PR-MPNN Qian et al. (2023), MolCLR Wang et al. (2022), HyperMol Cui et al. (2023), and MultiChem Moon & Rho (2025). For interpretability comparision, we employ Grad-CAM Selvaraju et al. (2017), GNNExplainer Ying et al. (2019), PGExplainer Luo et al. (2020), PGMExplainer Vu & Thai (2020), MatchExplainer Wu et al. (2023), and ReFine Wang et al. (2021) as baselines.

### 4.2 EXPERIMENTAL RESULTS

**Prediction Performance.** As shown in Table 1, when compared with the state-of-the-art models for molecular property prediction, our model achieves the best performance in the majority of scenarios.

| Dataset | BBBP | | BACE | | HIV | | AIDS | | MUTAG | | NCI1 | | BZR | |
|---------|------|------|------|------|------|------|------|------|-------|------|------|------|------|------|
| Method | Fid+ | Fid- | Fid+ | Fid- | Fid+ | Fid- | Fid+ | Fid- | Fid+ | Fid- | Fid+ | Fid- | Fid+ | Fid- |
| GradCAM | 30.1 | 32.5 | 28.3 | 30.5 | 26.3 | 22.6 | 6.5 | 16.6 | 26.3 | 15.8 | 25.1 | 26.1 | **9.7** | 12.9 |
| GNNExplainer | 29.6 | 30.5 | 30.3 | 32.8 | 23.1 | 27.3 | 5.0 | 11.1 | **15.8** | 25.2 | 23.4 | 27.4 | 22.6 | 25.8 |
| PGExplainer | 28.6 | 35.3 | 25.5 | 34.9 | 25.7 | 29.1 | 4.6 | 17.4 | 19.5 | 26.3 | 24.2 | 29.2 | 16.7 | 23.6 |
| PGMExplainer | 27.3 | 37.6 | 26.5 | 34.3 | 26.4 | 29.7 | 5.3 | 17.4 | 16.4 | 28.4 | 23.8 | 30.6 | 16.7 | 24.8 |
| MatchExplainer | 19.4 | 42.6 | 18.0 | 40.3 | 23.8 | 33.2 | 6.1 | 20.1 | **15.8** | 32.5 | 21.7 | 32.6 | 19.4 | 25.8 |
| ReFine | 20.3 | 44.3 | 18.6 | 41.6 | 24.2 | 32.4 | 4.0 | 18.6 | 26.3 | 43.6 | 22.9 | 35.2 | 19.4 | 29.0 |
| HYPERSEM (Ours) | **17.1** | **48.2** | **15.3** | **44.2** | **21.5** | **36.8** | **3.0** | **21.7** | 21.1 | **63.2** | **19.8** | **37.6** | 16.1 | **32.3** |

Table 2: Comparison of explanation performance in terms of Fid+ and Fid- scores across datasets. Detailed results are given in Appendix G.

Among all graph classification models, MultiChem performs well in most tasks, likely due to its ability to capture molecular information from multiple perspectives. In contrast, SMILES-based models perform poorly because they directly process SMILES strings, which do not provide an in-depth representation of molecular structures, leading to suboptimal results. In comparison with these models, HYPERSEM achieves nearly the best prediction performance across all tasks. This is due to our hypergraph model effectively captures the molecular structure, while the topological encoding mechanism retains crucial structural information. Additionally, the IB-guided hyperedge selection mechanism reduces irrelevant noise, thereby improving the model's prediction accuracy.

**Explanation Performance.** Table 2 summarizes the fidelity-based evaluation of explanations. (Detailed calculation is shown in Appendix E). When retaining key subgraphs (Fid+), our model achieves the lowest Fid+ scores on most datasets, particularly BBBP, BACE, HIV, and AIDS, indicating that the identified subgraphs are sufficient to preserve the original predictions. Although performance on MUTAG and BZR fluctuates, this may be attributed to their limited sample sizes,

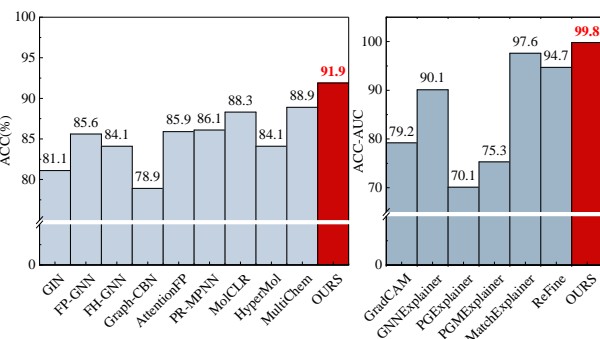

Figure 3: Prediction (left) and explanation (right) Results.

yet our approach still remains competitive. Meanwhile, when removing key subgraphs (Fid-), our model consistently achieves the best Fid- scores across all datasets, demonstrating that the excluded substructures are indeed necessary for the final predictions. These results validate that the explanatory subgraphs discovered by our method capture critical molecular substructures. By leveraging hyperedges to represent cooperative atoms and functional groups and employing IB-guided selection to highlight the most informative ones, our approach provides explanations that are both faithful and effective for molecular property prediction.

**Evaluation on Dataset with Ground-Truth Explanations.** In addition to the above benchmarks, we further evaluate our model on **Mutagenicity** Morris et al. (2020) dataset where ground-truth explanation substructures are available. Following prior work Luo et al. (2020), we adopt chemically meaningful functional groups, such as $NO_2$ and $NH_2$, as ground-truth explanations. As shown in Fig. 3, HYPERSEM achieves state-of-the-art performance on both the classification and explanation tasks under this ground-truth definition, with a remarkably high explanation AUC (99.8%).

**Ablation Study.** To validate the effectiveness of each module, we performed ablation experiments on different modules on 5 datasets. In terms of hypergraph construction, we ablated rule-based, local-based, and dynamic hyperedge construction. At the network level, we ablated topological encoding and mutual information constraints. The results of the ablation experiments are shown in Table 3 Acronyms used in the ablation are defined as follows: CGHC denotes Chemical Rule-Guided Molecular Hypergraph Construction; ANHC denotes Atom-Centered Neighborhood Molecular Hypergraph Construction; DHC denotes Dynamic Molecular Hypergraph Construction; SE denotes Structural Encoding; and IB refers to the Information Bottleneck constraint. It can be observed that the topological encoding module has a significant impact on the model's classification ability. Regarding the three different hypergraph construction methods, removing any of them resulted in a significant impact on both classification and interpretability performance, suggesting that the differ-

| Dataset | BBBP | | | BACE | | | HIV | | | MUTAG | | | AIDS | | |
|---|---|---|---|---|---|---|---|---|---|---|---|---|---|---|---|
| Varient | AUC | Fid+ | Fid- | AUC | Fid+ | Fid- | AUC | Fid+ | Fid- | ACC | Fid+ | Fid- | ACC | Fid+ | Fid- |
| w/o CGHC | 93.8 | 24.7 | 42.8 | 85.3 | 20.3 | 40.4 | 81.4 | 23.1 | 32.9 | 94.0 | 30.3 | 53.0 | 96.9 | 6.8 | 18.7 |
| w/o ANHC | 92.4 | 26.2 | 39.3 | 84.8 | 18.8 | 41.9 | 78.9 | 24.1 | 33.6 | 93.0 | 27.1 | 50.3 | 97.6 | 7.1 | 20.1 |
| w/o DHC | 95.6 | 20.4 | 45.7 | 87.6 | 19.4 | 39.1 | 80.1 | 23.6 | 32.1 | 95.7 | 27.1 | 55.7 | 96.5 | 6.3 | 19.1 |
| w/o SE | 87.5 | 24.5 | 44.2 | 84.2 | 20.1 | 38.5 | 76.1 | 22.4 | 34.5 | 94.0 | 23.8 | 52.5 | 93.1 | 5.2 | 18.9 |
| w/o IB | 95.3 | 22.6 | 42.5 | 88.1 | 22.4 | 37.3 | 82.4 | 26.3 | 30.1 | 98.9 | 25.4 | 48.7 | 98.3 | 11.2 | 17.6 |
| HYPERSEM | **97.2** | **17.1** | **48.2** | **90.4** | **15.3** | **44.2** | **84.6** | **21.5** | **36.8** | **99.7** | **21.1** | **63.2** | **99.7** | **3.0** | **21.7** |

Table 3: Results of ablation study. Detailed results are given in Appendix G.

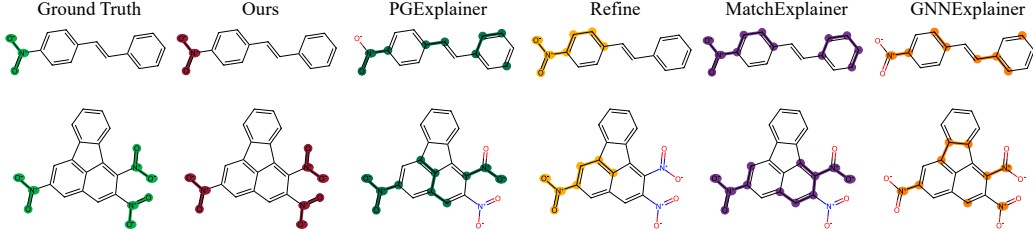

Figure 4: Visualization of molecular explanations produced by different methods.

ent strategies for hyperedge modeling effectively capture the atomic group structures in molecules. Furthermore, the ablation of the information bottleneck module shows that the mutual information mechanism, which guides the model to retain key hyperedges, not only effectively reduces the interference of noisy hyperedges but also helps the model focus on more representative hyperedges, thus improving classification and prediction performance.

**Visualization of Explanations.** To further validate the performance of our model in selecting interpretable subgraphs, we conducted a comparison of different explanation methods using MU-TAG dataset and chemical prior annotations as ground truth. As shown in Figure 4, both GN-NExplainer and PGExplainer produce numerous meaningless molecular fragments, such as segments on the benzene ring, and fail to capture key atoms in the nitro group. Although the Refine and MatchExplainer methods provide more complete subgraphs, they still fail to identify all the key atomic groups when faced with multiple critical groups, and are affected by interference from the benzene ring, resulting in subgraph parts overlapping with the benzene ring. Thanks to our hyperedge modeling mechanism for atomic groups, our model achieved the best results, accurately identifying all key nitro groups without being interfered by the benzene ring background.

**Visualization of Explanation Score Distribution.**
We further evaluate HYPERSEM on Mutagenicity dataset by analyzing the explanation scores assigned to different chemical groups. As shown in Figure 5, for functional groups such as *Alkyl* and *Halogen*, which are generally recognized as having little impact on mutagenicity, the scores assigned by HYPERSEM are mostly below 0.4. This indicates that our model is able to effectively exclude non-toxic substructures. In contrast, for functional groups that have been reported in the chemical literature Debnath et al. (1991); Yang et al. (2017) to exhibit strong

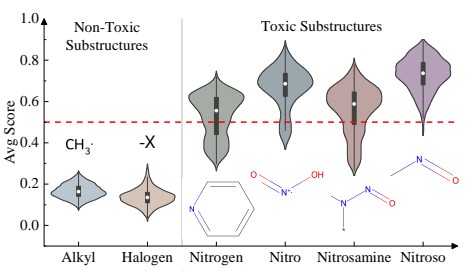

Figure 5: The distribution of contribution scores for non-toxic and toxic substructures.

mutagenic effects, such as *Nitro* and *Nitroso*, HYPERSEM assigns higher scores, with values around 0.7–0.8, suggesting that the model identifies them as critical contributors to molecular mutagenicity. For moderately toxic groups, such as *Nitrogen heterocycles* and *Nitrosamines*, the scores are around 0.6, reflecting that the model regards them as having a weaker yet still notable influence on molecular properties.

## 5 CONCLUSION

We proposed HYPERSEM, a novel hyperedge-based framework for explainable molecular property prediction. By introducing multi-view hypergraph modeling and a molecular structure-informed

hypergraph convolution, our approach captures cooperative effects of atom groups and functional group influences on molecular properties. Extensive experiments demonstrate that HYPERSEM achieves state-of-the-art performance in both prediction accuracy and explanation fidelity. This work highlights the potential of hypergraph-based explainability in advancing interpretable molecular modeling and opens avenues for future applications in molecular design and drug discovery.

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

# A ALGORITHM

To facilitate understanding of HYPERSEM 's workflow, we provide complete pseudo code for reference.

---

**Algorithm 1:** Multi-View Hypergraph Learning Method for Self-Explainable Molecular Property Prediction

---

**Input:** Molecular graph $G = (\mathcal{V}, \mathcal{E}_{\text{bond}})$; adjacency $\mathbf{A} \in \mathbb{R}^{N \times N}$; atom features $\mathbf{X} \in \mathbb{R}^{N \times d_{\text{in}}}$; $k$-hop radius $k$; distance cutoff $M$; layers $L$; dynamic hyperedges $H_{\text{dyn}}$; Gumbel temperature $\tau$ and noise $g_e \sim \text{Gumbel}(0,1)$; IB weight $\beta$ and prior $r(\mathbf{z}_{\text{mol}})$; learnable params $\Theta = \{\mathbf{W}_*, , \mathbf{a}_*, \mathbf{W}_{\text{dyn}}, \phi\}$.

**Output:** Prediction $\hat{y}$ and explanation $\mathcal{E}_{\text{explain}}$

**Stage 1: Multi-View Molecular Hypergraph Construction**

   // Chemical rule-guided molecular hyperedges

   $\mathcal{E}_{\text{rule}} = \{e_i \mid e_i = \text{GetAtoms}(f_i), f_i \in \text{BRICS}(\mathcal{G})\}$;

   // Atom-centered neighborhood molecular hyperedges

   $\mathcal{E}_{\text{khop}} = \{e_i \mid e_i = \mathcal{N}_k(v_i), \forall v_i \in \mathcal{V}\}$, where $\mathcal{N}_k(v_i) = \{v_j \in \mathcal{V} \mid \text{dist}(v_i, v_j) \leq k\}$;

   // Dynamic learnable hyperedges

   $\mathbf{s}_i = \mathbf{x}_i \mathbf{W}_{\text{dyn}}$;

   $h_{i,e} = \frac{\exp((s_{i,e} + g_e)/\tau)}{\sum_{e'} \exp((s_{i,e'} + g_{e'})/\tau)}$;

   $\mathcal{E}_{\text{dyn}} = \{e \mid \{h_{i,e}\}_{i=1}^N\}$;

   // Final hypergraph

   $\mathcal{H} = \{\mathcal{V}, \mathcal{E}_{\text{rule}} \cup \mathcal{E}_{\text{khop}} \cup \mathcal{E}_{\text{dyn}}\}$;

**Stage 2: Molecular Structure-Informed Molecular Hypergraph Convolution**

   // structural encoding

   **for** $v_i \in \mathcal{V}$ **do**

      **for** $v_j \in \mathcal{V}$ **do**

         $\text{dist}(v_i, v_j) = \min\{m \mid \mathbf{A}_{i,j}^m \neq 0\}$;

      **end**

   **end**

   **for** $e \in \mathcal{E}$ **do**

      **for** $v_i \in e$ **do**

         $N_{i,m}(e) = \sum_{v_j \in e} \mathbb{I}(\text{dist}(v_i, v_j) = m)$;

         $\mathbf{t}_{i,e} = [\, N_{i,-1}(e), N_{i,0}(e), N_{i,1}(e), \ldots, N_{i,M}(e)\,] \in \mathbb{R}^{M+1}$;

         $\mathbf{z}_{i,e} = \mathbf{W}_{nt}\mathbf{t}_{i,e} + \mathbf{b}_{nt}$;

      **end**

      $\mathbf{z}_e = \mathbf{W}_e \left( \frac{1}{|e|} \sum_{v_i \in e} \mathbf{t}_{i,e} \right) + \mathbf{b}_e$;

   **end**

   **for** $l = 0$ **to** $L - 1$ **do**

      // node to hyperedge message passing

      **for** $e \in \mathcal{E}$ **do**

         $\alpha_{i,e} = \frac{\exp(\text{LeakyReLU}(\mathbf{a}_1^\top [\mathbf{W}_1 \mathbf{x}_i^l \| \mathbf{W}_2 \mathbf{z}_{i,e}]))}{\sum_{v_k \in e} \exp(\text{LeakyReLU}(\mathbf{a}_1^\top [\mathbf{W}_1 \mathbf{x}_k^l \| \mathbf{W}_2 \mathbf{z}_{i,e}]))}$ ;

         $\mathbf{h}_e^l = \sigma\Big( \sum_{v_i \in e} \alpha_{i,e} \, \mathbf{W}_{\text{node}} \mathbf{x}_i^l \Big)$;

      **end**

      // hyperedge to node message passing

      **for** $v_i \in \mathcal{V}$ **do**

         $\beta_{e,i} = \frac{\exp(\text{LeakyReLU}(\mathbf{a}_2^\top [\mathbf{W}_3 \mathbf{h}_e^l \| \mathbf{W}_4 \mathbf{z}_e]))}{\sum_{e_k : v_i \in e_k} \exp(\text{LeakyReLU}(\mathbf{a}_2^\top [\mathbf{W}_3 \mathbf{h}_{e_k}^l \| \mathbf{W}_4 \mathbf{z}_{e_k}]))}$;

         $\mathbf{x}_i^{l+1} = \sigma\Big( \sum_{e : v_i \in e} \beta_{e,i} \, \mathbf{W}_{\text{edge}} \mathbf{h}_e^l \Big)$;

      **end**

   **end**

**Algorithm 1:** Multi-View Hypergraph Learning Method for Self-Explainable Molecular Property Prediction (continued)

---

**Stage 3: Information Bottleneck-Guided Self-Explanation**

    // Hyperedge scoring
    **for** $e \in \mathcal{E}$ **do**
    $\quad$ | $\quad s_e = \sigma(\mathbf{w}_s^\top \mathbf{h}_e^L + b_s)$ ;  $\quad\quad\quad$ // importance score of hyperedge $e$
    **end**
    // Hyperedge aggregation
    $\mathbf{z}_{\mathrm{mol}} = \sum_{e \in \mathcal{E}} \frac{s_e}{\sum_{e' \in \mathcal{E}} s_{e'}} \cdot \mathbf{h}_e^L$ ;
    // Prediction head
    $\hat{y} = f_\phi(\mathbf{z}_{\mathrm{mol}})$ ;  $\quad\quad\quad$ // final molecular property prediction
    // Information Bottleneck regularization
    $\mathcal{L}_{\mathrm{cls}} = \mathrm{CrossEntropy}(\hat{y}, y)$ ;
    $\mathcal{L}_{\mathrm{vib}} = \mathbb{E}_{p(\{\mathbf{h}_e^L\})}[\mathrm{KL}(p(\mathbf{z}_{\mathrm{mol}} | \{\mathbf{h}_e^L\}) \| r(\mathbf{z}_{\mathrm{mol}}))]$ ;
    $\mathcal{L}_{\mathrm{total}} = \mathcal{L}_{\mathrm{cls}} + \beta \cdot \mathcal{L}_{\mathrm{vib}}$ ;
    // Explanation output
    $\mathcal{E}_{\mathrm{explain}} = \{(e, s_e) \mid e \in \mathcal{E}\}$ ;  $\quad\quad\quad$ // rank hyperedges by importance
**return** $\hat{y}$, $\mathcal{E}_{explain}$

---

## B EFFICIENCY ANALYSIS

Our proposed HYPERSEM achieves a favorable balance between predictive performance and computational cost in both time and space. The dominant overhead arises from the molecular structure-informed Molecular Hypergraph Convolution. In particular, the multi-hop information in the structural encoding module can be precomputed for the static hypergraph to avoid repeated costs; for the dynamic hypergraph, the time complexity is $O(mE + N_{\mathrm{dyn}}M)$ and the space complexity is $O(mE)$, where $m$ and $M$ are constants and $E$ denotes the number of hyperedges.

For the molecular structure-informed Hypergraph Convolution, the time complexity is $O(NED)$ and the space complexity is $O((N + E)D)$. For the dynamic hyperedge selection module, the time and space complexities are both $O(ED)$. Summing up, the overall complexity of HYPERSEM is $O(NED + ED)$ in time and $O((N + E)D)$ in space (where $N$ is the number of nodes, $E$ the number of hyperedges, $D$ the feature dimension, and $N_{\mathrm{dyn}}$ the number of dynamic hyperedges).

To further substantiate this analysis, we report the inference runtime of HYPERSEM on the BBBP dataset, as summarized in Table 4.

| Dataset | Grad-CAM | GNNExplainer | PGExplainer | PGMExplainer | MatchExplainer | ReFine | HYPERSEM (Ours) |
|---|---|---|---|---|---|---|---|
| Inference Time(s) | 3.36±0.09 | 258.14±6.45 | 7.12±0.34 | 92.43±2.74 | 97.00±1.58 | 13.60±0.46 | 10.31±0.41 |

Table 4: Inference runtime in BBBP dataset

## C PARAMETER SENSITIVITY

To further illustrate how model performance varies with hyperparameter settings, we provide additional hyperparameter-sensitivity analyses on the BBBP, BACE, HIV, AIDS, MUTAG, NCI1, and BZR datasets. As shown in Table 5, we vary the number of dynamic hyperedges $N_{\mathrm{dyn}}$, the IB regularization strength $\beta$, the number of layers $L$, and the $k$ in the $k$-hop neighborhood used to construct the atom-centered hypergraph.

## D DATASET INTRODUCTION

**BBBP.** The Blood-Brain Barrier Penetration dataset classifies molecules as BBB-penetrant or non-penetrant. BBB permeability correlates with physicochemical descriptors such as molecular weight, lipophilicity (e.g., $\log P$), topological polar surface area (TPSA), and counts of H-bond donors/acceptors, and is routinely used in CNS lead selection and risk assessment.

| Param | Value | BBBP | | | BACE | | | MUTAG | | |
|---|---|---|---|---|---|---|---|---|---|---|
| | | AUC | Fid+ | Fid- | AUC | Fid+ | Fid- | ACC | Fid+ | Fid- |
| $N_{\mathrm{dyn}}$ | 8 | 96.2±0.9 | 18.8±3.7 | 45.9±2.6 | 87.1±2.4 | 17.1±2.3 | 42.1±1.8 | 96.3±2.6 | 23.1±3.0 | 60.3±1.7 |
| | 9 | 96.9±0.8 | 18.4±2.7 | 45.1±2.1 | 89.5±2.4 | 17.7±2.6 | 41.4±1.9 | **98.4±2.1** | **21.1±1.2** | **63.2±0.7** |
| | 10 | **97.2±2.0** | 17.1±3.6 | 48.2±2.4 | 90.3±1.4 | 16.7±3.1 | 42.2±2.4 | 97.4±2.1 | 23.5±3.5 | 61.4±2.5 |
| | 11 | 97.1±0.8 | 19.1±2.5 | 48.2±1.9 | **90.4±0.7** | **15.3±1.9** | **44.2±2.3** | 96.8±2.5 | 22.6±2.5 | 57.8±1.7 |
| | 12 | 95.1±1.3 | 19.3±2.9 | 47.1±2.3 | 88.4±1.9 | 16.5±3.2 | 43.2±2.2 | 95.7±1.3 | 23.8±2.9 | 57.3±1.6 |
| $\beta$ | 0.001 | 97.1±0.8 | 18.3±3.1 | 41.3±2.7 | 90.3±1.9 | 18.4±3.3 | 37.9±2.6 | 98.1±1.4 | 24.3±3.2 | 51.3±2.0 |
| | 0.01 | **97.2±2.0** | **17.1±3.6** | **48.2±2.4** | **90.4±0.7** | **15.3±1.9** | **44.2±2.3** | **98.4±2.1** | **21.1±1.2** | **63.2±0.7** |
| | 0.1 | 96.9±0.8 | 20.4±3.0 | 43.2±1.8 | 90.1±1.6 | 18.3±3.1 | 39.6±2.2 | 97.3±1.3 | 22.7±3.0 | 60.7±2.4 |
| | 1 | 94.3±2.2 | 19.3±2.9 | 39.1±2.4 | 87.7±2.4 | 17.3±3.2 | 35.9±2.6 | 94.9±2.3 | 23.6±3.1 | 57.2±2.2 |
| $L$ | 1 | 97.0±0.8 | 46.1±1.9 | 47.6±1.9 | **90.4±1.9** | 16.8±3.6 | 39.3±1.7 | 95.3±1.1 | 24.3±3.1 | 61.3±1.9 |
| | 2 | **97.2±2.0** | **17.1±3.6** | **48.2±2.4** | 89.4±1.9 | 15.7±3.2 | 42.1±2.0 | **98.4±2.1** | **21.1±1.2** | **63.2±0.7** |
| | 3 | 96.1±0.8 | 47.8±1.1 | 46.9±2.2 | **90.4±0.7** | **15.3±1.9** | **44.2±2.3** | 96.7±1.6 | 23.8±3.5 | 59.9±1.8 |
| | 4 | 94.3±1.9 | 48.4±1.3 | 47.3±1.6 | 88.7±2.4 | 15.9±3.5 | 44.3±1.8 | 96.3±1.2 | 23.9±3.4 | 58.0±1.5 |
| $k$ | 1 | 95.5±1.3 | 18.2±2.9 | 46.1±2.5 | 88.8±2.1 | 16.3±2.2 | 42.3±2.2 | **98.4±2.1** | **21.1±1.2** | **63.2±0.7** |
| | 2 | **97.2±2.0** | **17.1±3.6** | **48.2±2.4** | **90.4±0.7** | **15.3±1.9** | **44.2±2.3** | 96.3±1.3 | 23.2±3.5 | 60.6±1.7 |
| | 3 | 96.3±1.2 | 18.3±3.3 | 47.8±2.2 | 89.6±1.3 | 16.1±2.1 | 43.8±2.3 | 96.6±1.4 | 23.0±3.5 | 56.0±1.9 |
| | 4 | 95.8±1.3 | 19.2±3.1 | 48.3±1.6 | 86.1±1.9 | 17.1±3.0 | 42.3±2.9 | 95.1±1.5 | 24.8±3.8 | 57.3±2.3 |

Table 5: Comparison of models on BBBP, BACE, and MUTAG datasets across different parameters.

| Dataset | BBBP | BACE | HIV | AIDS | MUTAG | NCI1 | BZR | Mutagenicity |
|---|---|---|---|---|---|---|---|---|
| Graphs | 2039 | 1513 | 41128 | 2000 | 188 | 4110 | 405 | 4337 |
| Classes | 2 | 2 | 2 | 2 | 2 | 2 | 2 | 2 |
| Split | Scaffold | Scaffold | Scaffold | 10-Fold | 10-Fold | 10-Fold | 10-Fold | 10-Fold |
| Metric | ROC-AUC | ROC-AUC | ROC-AUC | ACC | ACC | ACC | ACC | ACC |

Table 6: Dataset description and metric description

**BACE.** The BACE dataset targets inhibition against $\beta$-secretase 1 (BACE-1), labeled as active vs. inactive. Because BACE-1 contributes to amyloid-$\beta$ production, the task reflects ligand–target binding via hydrogen-bonding networks, hydrophobic pocket fit, and $\pi$–$\pi$ stacking, informing lead optimization for Alzheimer's disease.

**HIV.** The HIV dataset classifies anti-HIV activity (active/inactive). Compounds may act on distinct viral targets—including reverse transcriptase, protease, and integrase—often featuring hydrophobic scaffolds and N-heterocycles; the task emphasizes identifying structural features predictive of suppressing viral replication in a multi-target setting.

**AIDS.** The AIDS dataset originates from the NCI AIDS antiviral screen (active/inactive), with broader scaffold diversity encompassing aromatic/heteroaromatic rings and charged groups. It probes the relationship between molecular properties and antiretroviral effect and serves as a complementary benchmark to the HIV dataset.

**MUTAG.** MUTAG is a binary mutagenicity benchmark on *Salmonella typhimurium* strains (mutagenic/non-mutagenic). Typical structural alerts include aromatic amines, nitro substituents, and electrophilic intermediates formed upon metabolic activation, highlighting structure–toxicity relationships.

**NCI1.** NCI1 derives from NCI anticancer screening and performs graph-level molecular classification (active/inactive) against human cancer cell lines. With diverse chemotypes (aromatic, heteroaromatic, and various substituted motifs), it stresses structure–activity relationships (SAR), cross-scaffold generalization, and robustness to weak signals.

**BZR.** The BZR dataset concerns binding activity to the Benzimidazole Receptor (active/inactive). Molecules commonly contain a benzimidazole ring system and substituted aryl/heteroaryl groups; the task reflects how substituent position, electronic effects, and steric factors modulate receptor affinity and pharmacological activity.

**Mutagenicity.** The Mutagenicity dataset provides a larger-scale benchmark for mutagenicity classification (mutagenic/non-mutagenic) spanning a wider chemical space (polycyclic aromatics, heterocycles, and substituted benzenes). Mutagenic risk is linked to formation of electrophilic metabolites and DNA reactivity, and the dataset is often paired with MUTAG to assess model robustness under greater structural diversity.

## E   FIDELITY CALCULATION

**Fidelity$^+$**: This metric evaluates the sufficiency of the identified important subgraph by measuring the discrepancy in classification performance between the model prediction on the complete graph $G$ and the explanatory subgraph $G_S$. It is formally defined as:

$$\text{Fidelity}^+ = \left| \text{Perf}(\{G_i\}_{i=1}^N) - \text{Perf}(\{G_{S_i}\}_{i=1}^N) \right| \tag{15}$$

where $\text{Perf}(\{G_i\}_{i=1}^N)$ denotes the classification performance metric (e.g., Accuracy, AUC) computed across all $N$ complete graphs, and $\text{Perf}(\{G_{S_i}\}_{i=1}^N)$ is the corresponding performance on the identified explanatory subgraphs. A **lower Fidelity$^+$** value indicates that the identified subgraph alone is sufficient to preserve the original classification performance.

**Fidelity$^-$**: This metric assesses the necessity of the identified important subgraph by measuring the discrepancy in classification performance between the model prediction on the complete graph $G$ and the complementary subgraph $G \setminus G_S$. It is formally defined as:

$$\text{Fidelity}^- = \left| \text{Perf}(\{G_i\}_{i=1}^N) - \text{Perf}(\{G_i \setminus G_{S_i}\}_{i=1}^N) \right| \tag{16}$$

where $\text{Perf}(\{G_i \setminus G_{S_i}\}_{i=1}^N)$ denotes the classification performance after removing the explanatory subgraphs. A **higher Fidelity$^-$** value indicates that excluding the identified subgraph significantly alters the classification performance, confirming its necessity.

## F   EXPERIMENTAL SETUP

We conducted our experiments on an NVIDIA RTX 3090 GPU. We set the initial learning rate to 0.001 with cosine-annealing decay, optimized with Adam, trained for up to 500 epochs, and employed early stopping.

Molecules are constructed from canonical SMILES using RDKit with standard sanitization (valence check, kekulization, charge normalization; invalid molecules are discarded). We build the molecular graph by adding an undirected edge for each covalent bond detected by RDKit. Atom features are initialized from RDKit descriptors (e.g., atomic number, degree, formal charge, aromaticity, hybridization, chirality, ring membership, and implicit hydrogen count). Unless otherwise stated, hydrogens are treated implicitly.

For fair and stable evaluation, on BBBP, BACE, and HIV we follow the dataset splits provided by the DeepChem [1] interface. Each method is run under 10 different random seeds, and we report the mean and standard deviation across runs. For AIDS, MUTAG, NCI1, and BZR, we adopt the 10-fold splits recommended by TUDataset [2] and likewise repeat each experiment with ten random seeds, reporting the mean and standard deviation. Our code will be released after this paper is accepted.

## G   DETAILED EXPERIMENTAL RESULTS

For those tables in the main text that omit standard deviations, we provide the complete versions (including standard deviations) below.

---

[1] https://deepchem.io/
[2] https://chrsmrrs.github.io/datasets/

| Dataset | BBBP | | BACE | | HIV | |
|---|---|---|---|---|---|---|
| Method | Fid+ | Fid- | Fid+ | Fid- | Fid+ | Fid- |
| GradCAM | 30.1±3.3 | 32.5±3.3 | 28.3±2.7 | 30.5±2.8 | 26.3±2.9 | 22.6±2.7 |
| GNNExplainer | 29.6±2.8 | 30.5±3.6 | 30.3±2.9 | 32.8±3.1 | 23.1±2.4 | 27.3±3.0 |
| PGExplainer | 28.6±2.5 | 35.3±3.0 | 25.5±2.4 | 34.9±3.2 | 25.7±2.6 | 29.1±2.8 |
| PGMExplainer | 27.3±2.3 | 37.6±2.7 | 26.5±2.3 | 34.3±3.0 | 26.4±2.5 | 29.7±2.9 |
| MatchExplainer | 19.4±1.9 | 42.6±2.5 | 18.0±1.8 | 40.3±2.7 | 23.8±2.2 | 33.2±2.4 |
| ReFine | 20.3±2.4 | 44.3±2.3 | 18.6±1.9 | 41.6±2.6 | 24.2±2.4 | 32.4±2.5 |
| OURS | **17.1±3.6** | **48.2±2.4** | **15.3±1.9** | **44.2±2.3** | **21.5±1.7** | **36.8±1.7** |

Table 7: Fid+ and Fid− scores across BBBP, BACE, and HIV datasets.

| Dataset | AIDS | | MUTAG | | NCI1 | | BZR | |
|---|---|---|---|---|---|---|---|---|
| Method | Fid+ | Fid- | Fid+ | Fid- | Fid+ | Fid- | Fid+ | Fid- |
| GradCAM | 6.5±1.4 | 16.6±1.6 | 26.3±3.0 | 15.8±2.4 | 25.1±2.2 | 26.1±1.2 | **9.7±1.8** | 12.9±2.1 |
| GNNExplainer | 5.0±0.9 | 11.1±1.5 | **15.8±1.9** | 25.2±2.8 | 23.4±1.9 | 27.4±1.3 | 22.6±2.6 | 25.8±2.7 |
| PGExplainer | 4.6±1.2 | 17.4±1.4 | 19.5±2.4 | 26.3±2.5 | 24.2±2.1 | 29.2±1.4 | 16.7±2.1 | 23.6±2.5 |
| PGMExplainer | 5.3±1.3 | 17.4±1.5 | 16.4±1.8 | 28.4±2.6 | 23.8±2.0 | 30.6±1.5 | 16.7±2.0 | 24.8±2.3 |
| MatchExplainer | 6.1±1.1 | 20.1±1.4 | **15.8±1.7** | 32.5±2.7 | 21.7±1.7 | 32.6±1.6 | 19.4±2.3 | 25.8±2.6 |
| ReFine | 4.0±1.0 | 18.6±1.3 | 26.3±2.3 | 43.6±2.9 | 22.9±2.0 | 35.2±1.7 | 19.4±2.1 | 29.0±2.4 |
| OURS | **3.0±0.2** | **21.7±0.9** | 21.1±1.2 | **63.2±0.7** | **19.8±1.4** | **37.6±0.4** | 16.1±1.3 | **32.3±2.2** |

Table 8: Fid+ and Fid− scores across AIDS, MUTAG, NCI1, and BZR datasets.

## H  VISUALIZATION OF DYNAMIC MOLECULAR HYPEREDGES

To evaluate the chemical validity of the dynamic hyperedges in our model, we visualized the dynamic hyperedges with the highest importance scores in several molecules from the MUTAG dataset. All these molecules' labels are mutagenic. The visualization results demonstrate that dynamic molecular hyperedges are capable of capturing functionally relevant atomic groups.

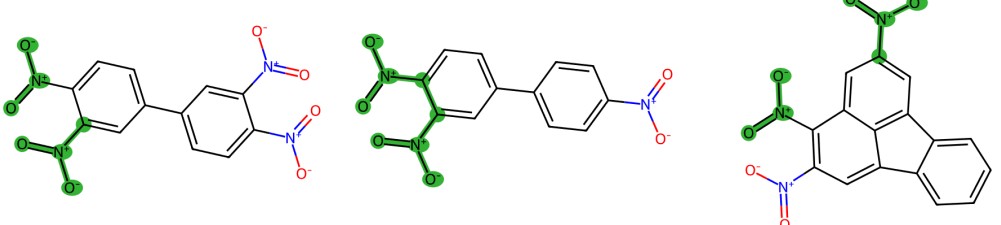

Figure 6: Visualization of the Most Important Dynamic Hyperedges on Some Molecules.

| Dataset | BBBP | | | BACE | | | HIV | | |
|---|---|---|---|---|---|---|---|---|---|
| Method | AUC | Fid+ | Fid- | AUC | Fid+ | Fid- | AUC | Fid+ | Fid- |
| w/o CGHC | 93.8±1.2 | 24.7±2.0 | 42.8±2.3 | 85.3±1.8 | 20.3±1.5 | 40.4±2.0 | 81.4±1.9 | 23.1±1.7 | 32.9±1.8 |
| w/o ANHC | 92.4±1.1 | 26.2±1.9 | 39.3±2.2 | 84.8±1.7 | 18.8±1.4 | 41.9±2.1 | 78.9±1.8 | 24.1±1.6 | 33.6±1.7 |
| w/o DHC | 95.6±1.0 | 20.4±1.7 | 45.7±2.4 | 87.6±1.6 | 19.4±1.3 | 39.1±2.2 | 80.1±1.8 | 23.6±1.5 | 32.1±1.6 |
| w/o GE | 87.5±1.3 | 24.5±2.1 | 44.2±2.5 | 84.2±1.8 | 20.1±1.6 | 38.5±2.3 | 76.1±2.0 | 22.4±1.9 | 34.5±2.2 |
| w/o IB | 95.3±1.2 | 22.6±1.8 | 42.5±2.3 | 88.1±1.5 | 22.4±1.7 | 37.3±2.1 | 82.4±1.9 | 26.3±1.6 | 30.1±1.8 |
| **OURS** | **97.2±2.0** | **17.1±3.6** | **48.2±2.4** | **90.4±0.7** | **15.3±1.9** | **44.2±2.3** | **84.6±1.6** | **21.5±1.7** | **36.8±1.7** |

Table 9: Ablation study with standard deviation on BBBP, BACE and HIV datasets

| Dataset | AIDS | | | MUTAG | | | NCI1 | | | BZR | | |
|---|---|---|---|---|---|---|---|---|---|---|---|---|
| Method | ACC | Fid+ | Fid- | ACC | Fid+ | Fid- | ACC | Fid+ | Fid- | ACC | Fid+ | Fid- |
| w/o CGHC | 96.9±1.0 | 6.8±0.8 | 18.7±1.0 | 94.0±2.6 | 30.3±1.5 | 53.0±4.2 | 83.0±1.2 | 23.0±1.6 | 36.7±1.7 | 84.0±1.1 | 18.6±1.6 | 27.9±1.8 |
| w/o ANHC | 97.6±1.2 | 7.1±1.0 | 20.1±1.2 | 93.0±6.3 | 27.1±0.4 | 50.3±3.7 | 80.2±1.3 | 22.3±1.5 | 36.3±1.6 | 83.1±1.2 | 16.5±1.4 | 29.6±2.1 |
| w/o DHC | 96.5±1.1 | 6.3±0.9 | 19.1±1.2 | 95.7±3.4 | 27.1±1.5 | 55.7±3.4 | 82.6±1.1 | 24.6±1.7 | 35.9±1.6 | 83.7±1.1 | 17.4±1.5 | 28.5±1.7 |
| w/o GE | 93.1±1.3 | 5.2±0.7 | 18.9±1.0 | 94.0±2.6 | 23.8±4.4 | 52.5±6.7 | 81.3±1.2 | 24.0±1.6 | 36.3±1.7 | 80.7±1.3 | 17.1±1.4 | 30.2±2.2 |
| w/o IB | 98.3±0.9 | 11.2±1.1 | 17.6±1.3 | 98.9±6.0 | 25.4±4.3 | 48.7±1.8 | 85.6±0.9 | 23.4±1.6 | 34.1±1.4 | 86.2±2.1 | 19.2±2.0 | 26.7±1.3 |
| **OURS** | **99.7±0.2** | **3.0±0.2** | **21.7±0.9** | **98.4±2.1** | **21.1±1.2** | **63.2±0.7** | **86.1±0.6** | **21.7±1.4** | **37.6±0.4** | **87.1±2.0** | **16.1±1.3** | **32.3±2.2** |

Table 10: Ablation study with standard deviation on AIDS, MUTAG, NCI1 and BZR datasets

