# OpenReview forum: "Self-explainable Molecular Property Prediction via Multi-view Hypergraph Learning"
_ICLR.cc/2026/Conference — Submitted to ICLR 2026_

### Official Review · Reviewer_LWZa · 2025-10-26

**Soundness:** 2
**Presentation:** 1
**Contribution:** 3
**Rating:** 2
**Confidence:** 4

**Summary:**

The paper introduces HYPERSEM, a novel multi-view hypergraph learning framework for molecular property prediction designed to provide self-explainability by modeling atomic groups as hyperedges. This model integrates chemical rules and data-driven signals using structure-informed hypergraph convolutions and an information bottleneck module for interpretable predictions and explanations. Experimental results demonstrate competitive accuracy and interpretability across seven benchmark datasets.

**Strengths:**

HYPERSEM achieves strong predictive performance, showing state-of-the-art results in benchmark comparisons

**Weaknesses:**

- The notation is dense and can be difficult to follow, with technical details scattered and some steps (especially implementation details) missing; this can impede reader understanding and reproducibility
- For interpretability benchmarks, the model is only compared against post-hoc explainers (e.g., GradCAM, GNNExplainer), not against other self-explainable approaches (PxGNN (Dai et al., 2025), PGIB (Seo et al., 2023), PiGNN (Ragno et al., 2021), GIP (Wang et al., 2024), KerGNNs (Feng et al., 2022), and IMPO (Ragno et al., 2025)). The chosen baselines are not always clearly detailed, and it's unclear which models the post-hoc explainers operate on.
- Table 1 conflates results for several methods but does not specify the metric used (ROC-AUC is only generally mentioned for the datasets in the appendix). Table 3 refers to "AUC" without stating which curve it is (likely ROC, but not specified), and whether it is for predictions, explanations, or another measure.
- Implementation transparency is lacking: there is no code or model artifact released, and crucial hyperparameters are under-specified, which hinders reproducibility and independent comparison. Hyperparameter sensitivity is summarized but not operationalized (Appendix C). Scientific standards require code availability for complex models like HYPERSEM.
- The ablation study uses unexplained acronyms (e.g., CGHC, ANHC, DHC, SE, IB) and does not detail them in the appendix or main text, limiting clarity for readers unfamiliar with the specific modules or experiment setup.
- The efficiency results (Table 4) indicate that HYPERSEM has higher inference time than GradCAM, which is surprising for a self-explainable approach, as explanations should ideally be simultaneous with predictions. The paper claims convolution complexity as the dominant cost, but does not offer implementation-level details or explicit justification.

**Questions:**

Could the notation and methodology steps be streamlined and clarified, possibly with a running example and more explicit definitions?​

Why are no self-explainable interpretability baselines considered, and which models are used for post-hoc interpretability comparisons?​

What exact metric is reported in Table 1 and 3? Is "AUC" always ROC-AUC, and is it used for explanation fidelity or prediction accuracy?​

Is the code or minimal reproducible artifact available/planned, with precise hyperparameter choices for all datasets and ablation experiments?​

What do the ablation acronyms stand for, and can these be defined for clarity?​

Why is HYPERSEM slower than GradCAM in inference, given that explanations are purportedly built-in? Can implementation specifics or bottlenecks be detailed?

---

> ### Author Response · Authors · 2025-11-22
>
> **Response Q1:** Thank you for the question. We provide a step-by-step pseudocode in Appendix A to clarify the full workflow and notation. In addition, we will release the complete code implementation and hyperparameter settings upon acceptance to ensure full reproducibility. We would be happy to address any further questions the reviewer may have.
>
> **Response Q2:** We have added more self-explainable methods on the BBBP and BACE dataset, including PGIB[1] and PiGNN[2], for your reference. For post-hoc methods, we consistently use GIN as the backbone. In the revised version, we will include additional self-explainable baselines in the experimental tables.
>
> | Dataset  | **BBBP** |      | **BACE** |      |
> | -------- | -------- | ---- | -------- | ---- |
> | Variant  | Fid+     | Fid- | Fid+     | Fid- |
> | PGIB     | 22.3     | 46.9 | 22.8     | 40.4 |
> | PiGNN    | 20.1     | 39.5 | 23.7     | 41.5 |
> | HyperSEM | 17.1     | 48.2 | 15.3     | 44.2 |
>
> **Response Q3:** The metrics reported in Table 1 follow the dataset descriptions provided in Appendix D. Specifically, we use ROC-AUC for BBBP, BACE, and HIV, and ACC (classification accuracy) for AIDS, MUTAG, NCI1, BZR, and Mutagenicity. The computation of fidelity (Fid) is described in Appendix E, where it is measured as the degree of performance degradation in the corresponding evaluation metric.
>
> **Response Q4:** We will release our code and hyperparameter settings after the paper is accepted to facilitate reproduction and follow-up work.
>
> **Response Q5:** Acronyms in the ablation are defined as follows: CGHC stands for Chemical Rule-Guided Molecular  Hypergraph Construction; ANHC stands for Atom-Centered Neighborhood  Molecular Hypergraph Construction; DHC stands for Dynamic Molecular Hypergraph Construction; SE stands for Structural Encoding; and IB denotes the Information Bottleneck constraint.
>
> **Response Q6:** For GradCAM as a post-hoc method, the additional computational cost for explanation mainly comes from a single backward pass. However, since molecular graphs are relatively small and the backbone GIN is fairly simple, the overhead introduced by this backward pass is actually not very large. In comparison, our model has a more complex design in the forward pass, including attention aggregation and a variational IB bottleneck, so the time it consumes is slightly longer than that of GradCAM.
>
> [1] Sangwoo Seo, Sungwon Kim, and Chanyoung Park. 2023. Interpretable prototype-based graph information bottleneck. In Proceedings of the 37th International Conference on Neural Information Processing Systems (NIPS '23). Curran Associates Inc., Red Hook, NY, USA, Article 3353, 76737–76748.
>
> [2] A. Ragno, B. La Rosa and R. Capobianco, "Prototype-based Interpretable Graph Neural Networks," in IEEE Transactions on Artificial Intelligence, 2022, doi: 10.1109/TAI.2022.3222618.

---

> > ### Comment · Reviewer_LWZa · 2025-11-25
> >
> > I appreciate the author's responses and the additional experiments. I am keen into increasing the score, however I still have the following concerns:
> > - I would like the code to be added now, as supplementary material. It does not have to be "clean" and "commented", but at least experiments should be replicable. Literature is full of too many approaches without code. And in this case the approach is far from being "straightforward" to reimplement.
> > - It is not clear how you obtain the explanatory subgraph, as attributions are usually "soft masks". How do you convert them into hard masks? I do not agree with Reviewer b3mD when he states that "BRICS fragments coincide with chemically meaningful motifs, explanations formed from these fragments inherently align with chemical intuition". The evaluation using fidelity cannot be biased by a "chemical intuition". However, what could bias the evaluation is removing an entire functional group with your approach and removing just a node with other SE-GNNs. How do you choose how many nodes you will remove?
> >
> > PS: you should not consider the forward pass in your time calculation, because it is needed also in the case of GradCAM. Either you consider the forward for all the approaches, or you don't for all of them. In case you don't, your approach yields explanations at prediction time? then the "explanation time" is 0, unless you have some calculation to transform explanations into masks, which goes back to the fact that some experimental details are still missing.

---

### Official Review · Reviewer_fcrj · 2025-10-27

**Soundness:** 2
**Presentation:** 2
**Contribution:** 2
**Rating:** 4
**Confidence:** 4

**Summary:**

The paper aims to improve the expressiveness and explainability of GNNs through multi-view hypergraph learning. The proposed method, HYPERSEM, constructs a multi-view hypergraph with three types of edges: (1) chemical rule-guided molecular hypergraph construction, (2) atom-centered neighborhood molecular hypergraph construction, and (3) dynamic molecular hypergraph construction. A convolutional network is proposed to encode structural information and pass messages between nodes and hyperedges. To provide graph explanations jointly with predictions, an information bottleneck-guided self-explanation module is introduced. Experiments on molecular property prediction and explanation tasks demonstrate the effectiveness of the proposed method.

**Strengths:**

1.	The paper focuses on inherently interpretable models that provide explanations while predicting molecular properties, which is an important and valuable direction to explore. The idea of using hyperedges to enhance interpretability is interesting.

2.	The methodology section is well organized, making the technical part relatively easy to follow.

3.	Experiments on both classification and explanation datasets demonstrate the effectiveness of the proposed method.

**Weaknesses:**

1. The research problem statement is not clearly defined. The paper claims that existing explanation methods are data-driven and ignore chemical rules. However, several prior works use fragments for explanation [1–4]. Although the proposed method introduces the BRICS decomposition method to generate hyperedges, it also employs other hyperedge generation methods that do not align with chemical rules. Moreover, the lack of evaluation by chemical practitioners makes it difficult to assess the interpretability of the final explanations.

2. The lack of comparison with related hypergraph-based methods obscures both the limitations of existing works and the novelty of the proposed approach [1, 5]. While some related work is mentioned in Sec. 2, the limitations of existing works and the novelties of the proposed work are not clearly stated.

3. Several technical details are missing, making the method difficult to evaluate. The meanings of $N_{i,m}(e), r(z_{\text{mol}})$, the calculation of $\mathcal{L}_{\text{vib}}$, the parameter setting of $\tau$, and the budget for final explanatory subgraphs are not clearly stated.

4. Some statements lack sufficient detail, making them difficult to understand. For example, in line 99, the authors state: “To alleviate the degradation of topological resolution commonly introduced by high-order hypergraph connections…” — but no supporting details or references are provided.

5. Some statements are ambiguous. For instance, in Eq. (3), the process of constructing hyperedges with soft assignments is unclear. It is not specified whether all edges belong to one hyperedge and how this process aligns with chemical knowledge, which raises interpretability concerns.

6. The criteria for selecting baseline methods are not clearly explained. In Sec. 2, the authors mention three methods that can provide interpretations jointly with predictions in an end-to-end manner, but these are not included in the comparisons.

7. Some modifications to compared methods are made without clear statements. For example, in Fig. 4, GNNExplainer and PGExplainer originally explain graphs through edge-level importance, but the visualizations show node-level importance.

[1] Yu, Z. and Gao, H., 2024. MAGE: Model-level graph neural networks explanations via motif-based graph generation. arXiv preprint arXiv:2405.12519.

[2] Kokate, A. and Fern, X., MOSE-GNN: A Motif-Based Self-Explaining Graph Neural Network for Molecular Property Prediction. In The Third Learning on Graphs Conference.

[3] Yu, Z. and Gao, H., 2022. Motifexplainer: a motif-based graph neural network explainer. arXiv preprint arXiv:2202.00519.

[4] Liu, X., Luo, D., Gao, W. and Liu, Y., 2025, July. 3dgraphx: Explaining 3d molecular graph models via incorporating chemical priors. In Proceedings of the 31st ACM SIGKDD Conference on Knowledge Discovery and Data Mining V. 1 (pp. 859-870).

[5] Bouritsas, G., Frasca, F., Zafeiriou, S. and Bronstein, M.M., 2022. Improving graph neural network expressivity via subgraph isomorphism counting. IEEE Transactions on Pattern Analysis and Machine Intelligence, 45(1), pp.657-668.

**Questions:**

1. Could the authors specify the computational cost of hyperedge generation in terms of both time and resources?

2. How are the first-level node embeddings $x_i^1$ derived?

3. Could the authors provide the chemical evaluation of the final explanation results? Additionally, could the authors report the proportions of the three types of hyperedges in the whole graphs and the explantory subgraphs?

4. In line 390, why are some methods using SMILES strings as input chosen, instead of those using the same type of molecular graph input for a fair comparison?

5. What metric is used for comparison in Table 1?

6. For visualization, could the authors clarify what the “chemical prior annotations” are and how they are obtained?

7. What are the budget settings for explanations? Why does the number of edges differ across methods in Fig. 4?

8. Which parameter settings are used for the experiments in Tables 1 and 2?

9. Could the authors define all mathematical symbols and their shapes from Eq. (5) to Eq. (9)?

10. Could the authors compare the results with substructure-based explanation methods [1–3]?

[1]. Lin, W., Lan, H. and Li, B., 2021, July. Generative causal explanations for graph neural networks. In International conference on machine learning (pp. 6666-6679). PMLR.

[2]. Bui, N., Nguyen, H.T., Nguyen, V.A. and Ying, R., 2024. Explaining graph neural networks via structure-aware interaction index. arXiv preprint arXiv:2405.14352.

[3]. Yuan, H., Yu, H., Wang, J., Li, K. and Ji, S., 2021, July. On explainability of graph neural networks via subgraph explorations. In International conference on machine learning (pp. 12241-12252). PMLR.

---

> ### Author Response · Authors · 2025-11-22
>
> **Response Q1:** The theoretical time complexity of the three hypergraph construction strategies is as follows: for the BRICS-based construction, the complexity is \(O(\#\text{bonds})\), since the BRICS fragmentation operates linearly over the molecular bonds; for the atom-centered \(k\)-hop neighborhoods, given a fixed \(k\), we calculate the \(k\)-hop relationships by precomputing and saving the \(k\)-th power of the adjacency matrix, with a complexity of approximately \(O(|E|)\); for the dynamic hyperedges, they are generated by a single linear transformation followed by Gumbel-Softmax, with a complexity of \(O(n \cdot N_{\text{dyn}})\), which is on the same order as a standard GNN layer (up to a constant factor related to the feature dimension).
>
> **Response Q2:** We construct standard molecular graph features for each atom (including atom type, valence electrons, degree, aromaticity, heteroatom indicators, chirality, etc.), and then map them to the hidden dimension used by our model via a single linear layer/MLP. In the revised version, we will state this explicitly in the methods section and provide a more complete list of atomic features in the Appendix.
>
> **Response Q3:** We follow the standard chemical evaluation protocol used in prior work [1,2], using known toxicophores as ground truth. Our visualizations show that the model recovers chemically meaningful motifs. We also report the proportions of the three hyperedge types on BBBP (19.4%, 56.0%, 24.6% vs. 42.1%, 26.5%, 31.4%), confirming that the model does not rely solely on BRICS.
>
> **Response Q4:** We select baselines that are strong and widely used models for molecular property prediction, including several SMILES-based methods. These models are standard and competitive on benchmarks such as MoleculeNet, allowing us to contextualize our predictive performance against established approaches.
>
> **Response Q5:** The metrics in Table 1 follow the dataset descriptions provided in the appendix: for the BBBP, BACE, and HIV datasets, we use ROC-AUC as the evaluation metric for classification performance; for the AIDS, MUTAG, NCI1, BZR, and Mutagenicity datasets, we use ACC (classification accuracy) as the evaluation metric.
>
> **Response Q6:** On the Mutagenicity task, we follow prior work [1,2] and use commonly adopted mutagenic functional groups / structural alerts as chemical prior annotations in explainability experiments.
>
> **Response Q7:**  We adopt a threshold-based strategy for hyperedge selection, so different molecular graphs may end up with different numbers of selected hyperedges, which in turn leads to different numbers of selected edges. The hyperparameter used for threshold selection will be released on GitHub after the paper is accepted.
>
> **Response Q8:** The parameter settings are listed in Appendix C, the bolded numbers are selected parameters. More detailed settings will be made available in our open-source GitHub repository after the paper is accepted.

---

> ### Author Response · Authors · 2025-11-22
>
> **Response Q9:** Thank you for your feedback. We have rephrased the following explanations of the formulas.
>
> In Eq(5), $N _ {i,m}(e)$ represents the number of nodes in the hyperedge $e$ that are at a minimum topological distance of $m$ from node $i$, which is a scalar. Here, $\mathbb{I}$ is an indicator function that takes the value 1 when $\text{dist}(v _ i, v _ j) = m$, and 0 otherwise. Eq(6) indicates that, for node $i$ in hyperedge $e$, the number of nodes at distances from $-1$ to $M$ hops from node $i$ are stored in a vector $\mathbf{t} _ {i,e}$, where $N _ {i,-1}(e)$ represents the number of nodes whose distance exceeds $M$ hops, which is set to -1. The vector $\mathbf{t} _ {i,e}$ has a length of $M+1$. Next, Eq(7) states that for the structural vector $\mathbf{z} _ {i,e}$ of node $i$ in hyperedge $e$, we learn it through a linear layer (with weight matrix $\mathbf{W} _ {nt} \in \mathbb{R}^{H \times (M+1)}$ and bias term $\mathbf{b} _ {nt}$). The structural vector $\mathbf{z} _ e$ for hyperedge $e$ is learned by averaging the structural vectors of the nodes in the hyperedge and passing it through another linear layer (with weight matrix $\mathbf{W} _ e \in \mathbb{R}^{H \times (M+1)}$ and bias term $\mathbf{b} _ e$), where $H$ is the hidden layer dimension, which is manually set.
>
> For Eq(8) and Eq(9), we describe the attention-based message passing function. Specifically, we first concatenate the node's structural vector with its feature vector and use the attention mechanism to aggregate information to the hyperedge, thus generating the hyperedge feature. In Eq(8), $\alpha _ {i,e}$ represents the attention coefficient for node $v _ i$ with respect to hyperedge $e$. This coefficient is calculated by computing the attention score of the concatenated vector of node feature $\mathbf{x} _ i^l$ and node structural vector $\mathbf{z} _ {i,e}$, followed by applying a LeakyReLU activation function and a softmax operation. $\mathbf{h} _ e^l$ is the hyperedge feature, which is calculated by aggregating the node features of all nodes connected to hyperedge $e$, with the weight being the corresponding attention coefficient $\alpha _ {i,e}$.
>
> In Eq(9), $\beta_{e,i}$ is the attention coefficient of hyperedge $e$ with respect to node $v _ i$. This coefficient is calculated by computing the attention score of the concatenated vector of hyperedge feature $\mathbf{h} _ e^l$ and hyperedge structural vector $\mathbf{z} _ e$, followed by a LeakyReLU activation function and softmax operation. $\mathbf{x} _ i^{l+1}$ is the feature of node $v _ i$ in the next layer, which is calculated by aggregating the weighted features of all hyperedges containing node $v _ i$, with the weight being the corresponding attention coefficient $\beta _ {e,i}$.
>
> These steps, based on the attention mechanism, allow the hyperedge features to effectively aggregate information from different nodes and feedback that information to the nodes. This enables nodes to update their representations based on the global information from the hyperedges, ultimately resulting in richer node representations.
>
> **Response Q10:** Comparison results are listed as follows:
>
> | Dataset      | **BBBP** |      | **BACE** |      |
> | ------------ | -------- | ---- | -------- | ---- |
> |              | Fid+     | Fid- | Fid+     | Fid- |
> | Gem[3]       | 23.6     | 44.2 | 22.8     | 39.4 |
> | MAGE[4]      | 20.7     | 39.3 | 20.1     | 41.6 |
> | SubgraphX[5] | 18.3     | 26.5 | 24.7     | 34.8 |
> | HyperSEM     | 17.1     | 48.2 | 15.3     | 44.2 |
>
> [1] Jiahua Rao, Hanjing Lin, Jiancong Xie, Zhen Wang, Shuangjia Zheng, and Yuedong Yang. 2025. Incorporating Retrieval-based Causal Learning with Information Bottlenecks for Interpretable Molecular Graph Learning. In Proceedings of the 31st ACM SIGKDD Conference on Knowledge Discovery and Data Mining V.2 (KDD '25).
>
> [2]  Luo, D., Cheng, W., Xu, D., Yu, W., Zong, B., Chen, H., and Zhang, X. 2020. Parameterized Explainer for Graph Neural Network. In Advances in Neural Information Processing Systems 33 (NeurIPS 2020).
>
> [3] Lin, W., Lan, H. and Li, B., 2021, July. Generative causal explanations for graph neural networks. In International conference on machine learning (pp. 6666-6679). PMLR.
>
> [4] Bui, N., Nguyen, H.T., Nguyen, V.A. and Ying, R., 2024. Explaining graph neural networks via structure-aware interaction index. arXiv preprint arXiv:2405.14352.
>
> [5] Yuan, H., Yu, H., Wang, J., Li, K. and Ji, S., 2021, July. On explainability of graph neural networks via subgraph explorations. In International conference on machine learning (pp. 12241-12252). PMLR.

---

> > ### Comment · Reviewer_fcrj · 2025-11-26
> >
> > Thank the authors for the detailed responses and the revised version of the paper. However, after carefully reading both the rebuttal and the updated paper, several concerns remain:
> >
> > 1. The authors provide theoretical time complexity in the response and runtime results in Table 4 (Appendix). For a fair comparison, all computational costs should be included, including preprocessing and post-processing. Since the three hypergraph construction methods may improve performance, a complete and detailed analysis of runtime and memory usage is needed to understand the trade-offs among the methods.
> >
> > 2. The explanation regarding the construction of initial embeddings remains unclear. The authors should clarify how atom-level features (e.g., atom type, valence electrons, degree, aromaticity, heteroatom indicators, chirality, etc.) are obtained for the experimental datasets and how these features are processed via an MLP.
> >
> > 3. The references provided cover ground-truth explanations only for the MUTAG dataset. Additional supporting evidence should be provided for other datasets. The response regarding BBBP is also unclear—what does “vs.” mean in this context? Moreover, it is unclear whether the Neighborhood-based and Dynamic methods can truly extract chemically meaningful substructures; please elaborate on the underlying reasons.
> >
> > 4. More details should be provided on the criteria used to select the baselines. The authors may refer to W2.
> >
> > 5. Although some clarifications were provided, it would be better to place the evaluation metrics in the experimental section and explain why different but similar datasets use different metrics.
> >
> > 6. For Q7, it remains unclear why PGMExplainer is not visualized in Figure 4. Which budget was used to select subgraphs for comparison? The ground-truth explanations are derived from PGExplainer, so why is it difficult to find the ground truth for PGExplainer? Additional visualizations for other datasets would strengthen the claim.
> >
> > 7. For Q8, the authors should clarify how the parameters for baseline methods were tuned. Providing parameter settings of the proposed method for all datasets in the Appendix would improve reproducibility. An ablation study illustrating parameter sensitivity is also recommended.
> >
> > 8. In Table 6, the number of MUTAG graphs differs from that presented in PGExplainer. The authors should explain this discrepancy.
> >
> > 9. For Q9, further clarification is needed regarding why the method uses attention-based message passing. Where are the definitions of $Q, K$, and $V$ in the equations for the attention?
> >
> > 10. For Eq. 6, what is the setting of $M$ in the experimental part?
> >
> > 10. More details should be provided for the experimental settings in the response to Q10.
> >
> >
> >
> >
> > Since most of the concerns still exist, it is better for the authors to clarify the above issues, as well as the comments raised in the Weaknesses section, to improve the clarity and quality of the paper.

---

### Official Review · Reviewer_b3mD · 2025-10-28

**Soundness:** 3
**Presentation:** 3
**Contribution:** 2
**Rating:** 4
**Confidence:** 4

**Summary:**

This paper introduces HYPERSEM, a self-explainable molecular property prediction framework that models molecules as multi-view hypergraphs to capture cooperative effects among atom groups while integrating chemical rules through BRICS fragmentation. The method constructs hyperedges from chemical rules, atom-centered neighborhoods, and dynamically learned connections, followed by a structure-informed hypergraph convolution that preserves molecular topology. An information bottleneck-guided module jointly produces predictions and interpretable substructure-level explanations by identifying informative hyperedges. Experiments on multiple molecular benchmarks demonstrate improved predictive accuracy and chemically meaningful explanations compared with existing GNN-based explainers.

**Strengths:**

1. The paper presents a clear framework (HYPERSEM) that integrates multi-view hypergraph modeling and information bottleneck-guided self-explanation for molecular property prediction.

2. The approach achieves competitive results across multiple molecular benchmarks and provides detailed ablation and visualization studies.

3. The idea of combining BRICS-based chemical rule-driven hyperedges with dynamic data-driven hyperedges offers a structured way to infuse domain knowledge into GNN interpretability.

4. The paper is well-presented and the experiments are comprehensive.

**Weaknesses:**

1. Motivational insufficiency. The two claimed limitations motivating this work, (1) neglect of atom groups, and (2) ignorance of chemical rules, are not comprehensively validated. Many prior works already address functional group or motif-level reasoning within molecular GNNs, including hyperstructure-based GNNs (e.g., DHKH [1]) and motif-based explanation frameworks (e.g. MOSE-GNN[2] ). The paper should provide a more rigorous comparison to these approaches to substantiate novelty.

2. Limited novelty of BRICS integration. The use of the BRICS fragmentation algorithm for constructing chemically meaningful motifs is not new. Similar BRICS-based or functional-group decomposition strategies have appeared in previous molecular hypergraph and interpretable-GNN literature (e.g., MAGE[3]). The contribution here mainly lies in combining known strategies rather than introducing a fundamentally new mechanism.

3. Technical opacity. The explanation generation process lacks precise details. The paper mentions hyperedge importance scoring but omits the selection rule, whether Top-K substructures or thresholding is applied. For fair comparison, it should be clarified how the number of selected nodes or edges is normalized across different explainers.

4. Bias in explanation quality. The interpretability advantage may be partly induced by the BRICS-based hyperedge construction. Since BRICS fragments coincide with chemically meaningful motifs, explanations formed from these fragments inherently align with chemical intuition, creating an inductive bias rather than true learned interpretability. This limitation should be acknowledged and discussed explicitly. Moreover, there are a lot of molecules that BRICS can not work for, how to tackle this problem?

5. A small typo: Page 6 line 320: "$z_{mol}$" appears incorrectly.

[1] Kang, Xiaojun, et al. "Dynamic hypergraph neural networks based on key hyperedges." Information Sciences 616 (2022): 37-51.
[2] Kokate, Apurva, and Xiaoli Fern. "MOSE-GNN: A Motif-Based Self-Explaining Graph Neural Network for Molecular Property Prediction." The Third Learning on Graphs Conference.
[3] Yu, Zhaoning, and Hongyang Gao. "MAGE: Model-level graph neural networks explanations via motif-based graph generation." arXiv preprint arXiv:2405.12519 (2024).

**Questions:**

See weakness.

I am willing to modify my score after further revision and discussion.

---

> ### Author Response · Authors · 2025-11-22
>
> **Response W1:** Thank you for the thoughtful comment. We appreciate the reviewer’s mention of prior works such as DHKH[1] and MOSE-GNN[2]. We would like to clarify that, to the best of our knowledge, these methods do not report quantitative evaluation.  The key contribution of our work lies in leveraging hypergraphs to explicitly capture multi-atom group structures that influence molecular properties, allowing us to jointly enhance predictive accuracy and interpretability within a unified framework. Our model not only delivers accurate molecular property predictions but also produces endogenous, task-relevant explanations, which makes it genuinely self-explainable.
>
> **Response W2:** We thank the reviewer for the insightful comment. We agree that BRICS is not novel and has been used in prior interpretable molecular GNNs (e.g., MAGE[3]). However, our method does not rely on BRICS alone. The novelty lies in integrating BRICS into a unified multi-view, self-explainable hypergraph framework, which fundamentally differs from prior BRICS-based approaches. Our model combines three complementary types of hyperedges to learn task-specific substructures beyond predefined motifs. Moreover, unlike post-hoc explainers such as MAGE, our explanations are jointly optimized with prediction, ensuring that the explanatory structure directly influences the model’s decision process. This also avoids the substantial additional training and sampling overhead inherent to post-hoc generative explainers like MAGE.
>
> **Response W3:** We appreciate the reviewer’s comment. Since molecule sizes and the number of hyperedges vary across samples, we use a threshold-based selection instead of a fixed Top-K, which would otherwise include irrelevant hyperedges for small molecules or exclude important ones for larger molecules.
>
> **Response W4:** We thank the reviewer for the comment. BRICS is used only as a non–task-specific chemical prior and does not encode any label information. Our model is not constrained to follow BRICS fragments. The IB objective allows it to autonomously select the most predictive hyperedges, so the explanations reflect the model’s reasoning rather than the prior. Moreover, our framework does not rely solely on BRICS: we also include atom-centered and data-driven dynamic hyperedges, which ensures coverage and robustness for molecules where BRICS fragmentation is incomplete or not applicable. In addition, our model architecture is flexible and can incorporate alternative decomposition rules, which we consider a promising direction for future work.
>
> **Response W5:** Thanks for your careful reading, we will fix this typo error in revised version.
>
> [1] Kang, Xiaojun, et al. "Dynamic hypergraph neural networks based on key hyperedges." Information Sciences 616 (2022): 37-51.
>
> [2] Kokate, Apurva, and Xiaoli Fern. "MOSE-GNN: A Motif-Based Self-Explaining Graph Neural Network for Molecular Property Prediction." The Third Learning on Graphs Conference.
>
> [3] Yu, Zhaoning, and Hongyang Gao. "MAGE: Model-level graph neural networks explanations via motif-based graph generation." arXiv preprint arXiv:2405.12519 (2024).

---

> > ### Comment · Reviewer_b3mD · 2025-11-25
> > **Response to Authors' Rebuttal**
> >
> > Thank you for the detailed rebuttal and clarifications. Some of my questions are solved. However, some of my main concerns still exist.
> >
> > 1. While I appreciate the distinctions the authors draw, the overall idea of using hypergraph structures to support molecular explanation is already well explored; applying the same principle to property prediction does not introduce a fundamentally new conceptual advance, but rather a different setting of an existing idea.
> >
> > 2. Although the paper highlights “simultaneous prediction and explanation,” this capability arises naturally once any hypergraph-based GNN incorporates an explanatory objective. The ablation study result is a little weird to me. Removing the IB constraint should enable the GNN to only focus on prediction. However, this leads to even worse prediction results. Could the authors explain this phenomenon?
> >
> > 3. The fairness issue in explanation evaluation is still open: threshold-based selection produces explanation subgraphs of variable size, whereas several baselines rely on fixed top-K, and the proposed explanation does not clarify how the comparison is normalized across explainers to ensure equivalent explanation scales. This is also shown in Figure 4.
> >
> > Because these concerns remain insufficiently addressed, my overall assessment remains the same.

---

### Official Review · Reviewer_AC1y · 2025-10-29

**Soundness:** 1
**Presentation:** 3
**Contribution:** 2
**Rating:** 4
**Confidence:** 3

**Summary:**

This paper introduces a new framework that models molecules as hypergraphs instead of simple atom–bond graphs to better capture atomic group effects (e.g. functional groups) and provide interpretable predictions. It constructs three complementary types of hyperedges, rule-based (using BRICS chemical fragmentation), atom-centered local neighborhoods, and dynamic data-driven hyperedges, to encode both chemical priors and learned structural relations. A molecular structure-informed hypergraph convolution with shortest-path-based structural encoding enables bidirectional message passing between atoms and atom groups, while an information-bottleneck-guided self-explanation module selects the most informative hyperedges to yield both accurate predictions and chemically meaningful explanations.

**Strengths:**

1. Their methods yields good results based on what the authors report

2. The paper is roughly easy to follow and well-written

3. The use of a rule-based fragmentation (e.g., BRICS) is interesting. It is a way to integrate known chemical knowledge.

**Weaknesses:**

1. The baselines for interpretability seem very old. The newest one is from 2023, while most are even from 2020 or pre 2020.

2. The motivation of this paper seems conceptually problematic. The authors argue that existing methods suffer from two main drawbacks: (1) neglect of atom groups and (2) ignorance of chemical rules. However, the fundamental goal of an explanation method is to reveal how a neural network makes its decisions and to assess whether those decisions make sense or align with human knowledge. If human knowledge, such as chemical rules and functional group structures, is already explicitly incorporated into the model or explanation mechanism, then it becomes unclear what the method is actually explaining. In such a case, the explanation may simply reflect the prior knowledge injected into the system, rather than providing genuine insight into the model’s reasoning process.

3. In addition to the motivation, without the consideration of model explanation, there are already existing works on hyperstructure-based GNNs.

4. The experimental comparison can be unfair. The reported interpretability gains may partly stem from the use of BRICS-based hyperedge construction rather than genuine model reasoning. Because BRICS fragments naturally correspond to chemically meaningful motifs, explanations derived from these predefined fragments are predisposed to align with chemical intuition, introducing a strong inductive bias rather than demonstrating truly learned interpretability. This potential bias should be explicitly acknowledged and discussed. Furthermore, many molecules cannot be effectively decomposed by BRICS, raising concerns about how the method would generalize to such cases and what strategies could be used to address this limitation.

**Questions:**

1. The explanation mechanism relies on minimizing the information bottleneck loss, yet the resulting substructures that achieve this objective are not necessarily chemically meaningful. The authors could test this by exhaustively or combinatorially evaluating finite substructure combinations to verify whether those minimizing the IB loss correspond to valid and interpretable chemical groups.

2. The paper positions HYPERSEM as self-explainable. In what sense is it “self-explainable” beyond simply outputting attention scores or IB-derived importance weights?

---

> ### Author Response · Authors · 2025-11-22
>
> **Response W1**: Thanks for your insightful advice, we chose these methods because they have solid and high reputation in the explanatory direction. We further compare several new methods, including MAGE(2024)[1] and GraphEXT(2025)[2] in the following tables:
>
> | Dataset     | **BBBP** |      | **BACE** |      |
> | ----------- | -------- | ---- | -------- | ---- |
> | Variant     | Fid+     | Fid- | Fid+     | Fid- |
> | MAGE[1]     | 22.6     | 40.8 | 23.7     | 40.4 |
> | GraphEXT[2] | 20.2     | 41.1 | 16.7     | 42.8 |
> | HyperSEM    | 17.1     | 48.2 | 15.3     | 44.2 |
>
> **Response W2**: We believe that the reviewer’s characterization of “the goal of explanation methods being to assess whether the model’s decisions are reasonable or align with human knowledge” highlights an important aspect of interpretability, but may not fully capture the broader objectives of interpretable models. Beyond revealing the internal decision-making mechanism of the model, another key goal of interpretability is to ensure that, when the model is capable of making accurate predictions, its explanations remain as consistent as possible with human domain knowledge, thereby improving comprehensibility and trustworthiness. The BRICS chemical rules we incorporate serve only as publicly available and widely accepted chemical priors for constructing candidate structural units, rather than providing any task-specific answers. Which structures are emphasized, how they are combined, and how they contribute to the prediction are all determined automatically by the model through learning from data. Therefore, our method does not forcibly inject human knowledge into the explanations; instead, it supplies a reasonable structured prior that helps the model produce explanations that are more aligned with human understanding while remaining faithful to the model’s own reasoning process.
>
> **Response W3:** We agree that there already exist works on hyperstructure-based GNNs. However, we would like to clarify that most existing hyperstructure-based GNNs primarily focus on improving classification performance, while their interpretability is largely overlooked. To the best of our knowledge, our work is the first to directly treat hyperedges themselves as interpretable units, which distinguishes our approach from prior methods.
>
> **Response W4:** We use BRICS as a prior because it is a publicly available and widely recognized chemical rule set, which contains no task-specific or label-related information. Therefore, it does not directly bias the explanations toward any particular label. More importantly, our interpretability training does not enforce the model’s explanations to align with BRICS fragments. Instead, through an information bottleneck constraint, the model is encouraged to autonomously learn which structural units are most informative and most supportive of its predictions while maintaining strong classification performance. In other words, the explanations are determined by the model’s actual reasoning process rather than injected from the prior. Meanwhile, our method does not rely solely on BRICS-based hyperedge construction; we also incorporate two additional, independent hyperedge construction strategies to ensure effectiveness and generalizability across different types of molecular structures.
>
> **Response Q1:** We appreciate the reviewer’s suggestion. To further support the chemical meaningfulness of the learned substructures, we have additionally visualized several dynamically hyperedges in the Appendix H. These examples show that the substructures identified under the IB objective align well with chemically interpretable motifs.
>
> **Response Q2:** In our setting, a model is considered self-explainable if it provides the basis for its prediction simultaneously with the prediction itself. HYPERSEM outputs not only the molecular property prediction but also the key structural units that contribute to this prediction. Therefore, it meets the criterion of being self-explainable.
>
> [1] Yu, Zhaoning, and Hongyang Gao. "MAGE: Model-level graph neural networks explanations via motif-based graph generation." *arXiv preprint arXiv:2405.12519* (2024).
>
> [2] Lijun Wu, Dong Hao, and Zhiyi Fan. 2025. Explainable graph neural networks via structural externalities. In Proceedings of the Thirty-Fourth International Joint Conference on Artificial Intelligence (IJCAI '25). Article 66, 583–591. https://doi.org/10.24963/ijcai.2025/66

---

> ### Comment · Reviewer_AC1y · 2025-11-24
>
> Thank you for your response.
>
> - W2:
>     - What I am suggesting is that you claim a good explanation method should explain using functional/atomic groups (limitation 1) or chemical rules (limitation 2). Then you use fragments as the building blocks for representing a molecular graph. This gives you a natural advantage under your own definition of what constitutes a “good explanation".
>     - I could not find where in your paper an explanation is even defined. I assumed you follow the definition in [1].
>
> [1] GNNExplainer: Generating Explanations for Graph Neural Networks
>
> - W3:
>     - I do not find this to be novel. In those works, they simply do not concern themselves with explainability and instead focus on model performance. If they were to apply GNN explanation methods to their hyperstructure-based GNNs, those hyperstructures would naturally emerge as the “interpretable units" as well.
>
> - Q1:
>    - Ignore my Q1. For your particular test cases, there are no other substructures that can yield better Perf. I had a misunderstanding.

---

> > ### Author Response · Authors · 2025-11-25
> >
> > Thank you for your response.
> >
> > **Response W2:** A “good explanation” in our work is fundamentally fidelity-based and model-behavior-based, following the same standard definition as GNNExplainer, i.e., identifying substructures that are sufficient and necessary for the model’s prediction. This is exactly what our Fid⁺/Fid⁻ metrics evaluate.
> >
> > At the same time, because our model is self-explainable, meaning that it outputs the explanation jointly with the prediction, we expect its explanations to be usable and cognitively meaningful to human experts. For this reason, we introduce chemical priors (e.g., BRICS) not as the definition of an explanation, but merely as a way to regularize the candidate structural units over which explanations are learned.
> >
> > Importantly, the explanation itself is entirely determined by the model’s learned behavior through the IB-guided objective and not by any predefined chemical rule. Many high-importance hyperedges in our results are dynamically learned rather than BRICS-derived, illustrating this point.
> >
> > To avoid confusion, we will add an explicit definition of “explanation” in the revised version, aligned with the model-behavior-based definition commonly adopted in explainable GNN literature.
> >
> > **Response W3:** We agree that hypergraph GNNs can be combined with post-hoc explanation methods. However, our contribution is not about merely treating hyperedges as interpretable units.
> >
> > Our novelty lies in introducing the first end-to-end self-explainable hypergraph framework for molecular property prediction, where explanation is integrated into the model architecture rather than added post hoc. Existing hypergraph GNNs are designed primarily for predictive performance and do not include mechanisms explicitly tailored for interpretability.
> >
> > In contrast, HYPERSEM incorporates a dedicated hyperedge-level explanation pipeline, including multi-view hyperedge construction, structure-informed message passing, and an information-bottleneck–guided importance selection. This design allows the model to achieve state-of-the-art performance on both prediction and explanation. Therefore, our novelty is architectural and methodological: we build a hypergraph model for the purpose of explanation, rather than applying an explainer on top of an existing hypergraph model.

---

> > > ### Comment · Reviewer_AC1y · 2025-11-28
> > >
> > > Thanks to the authors for their responses.
> > >
> > > - W2 To avoid confusion, we will add an explicit definition of “explanation” in the revised version, aligned with the model-behavior-based definition commonly adopted in explainable GNN literature.
> > >   - I think you should add the definition to make your paper self-contained. I am familiar with the definition, so I understand how your quantitative performance is measured. What I am suggesting is that your method has a natural advantage given your motivation. This point is not related to your performance itself.
> > >
> > > - W3 We agree that hypergraph GNNs can be combined with post-hoc explanation methods. However, our contribution is not about merely treating hyperedges as interpretable units.
> > >   - My response was based on your earlier statement: "To the best of our knowledge, our work is the first to directly treat hyperedges themselves as interpretable units, which distinguishes our approach from prior methods."
> > >   - From your new response, I assume you agree with my interpretation of your previous statement and that your admit this is not very novel and not a direct contribution.
> > >   - I agree that your method is “self-explainable.” While I think there is some contribution there, I am not fully convinced of its practical usefulness. Could you provide several clear downsides of post-hoc explanation methods compared with your approach?

---

### Official Review · Reviewer_nzLU · 2025-11-01

**Soundness:** 3
**Presentation:** 3
**Contribution:** 2
**Rating:** 6
**Confidence:** 4

**Summary:**

This paper proposes a multi-view hypergraph learning framework for self-explainable molecular property prediction. The key innovation is using hyperedges to model atomic groups and functional groups, moving beyond traditional node/edge-based explanations. The method combines three types of hyperedges: chemical rule-guided (BRICS), atom-centered k-hop neighborhoods, and dynamically learned hyperedges. A molecular structure-informed hypergraph convolution and information bottleneck-guided self-explanation module are designed to jointly generate predictions and explanations.

**Strengths:**

The paper clearly identifies two critical limitations of existing explainable GNN methods for molecular property prediction: 1) neglect of cooperative effects among atomic groups, and 2) ignorance of chemical domain knowledge.

**Weaknesses:**

- Is there a potential issue of K-hop Neighborhood Redundancy? For example, in the k-hop hyperedges (Eq. 2), for a molecule with N atoms, you create N hyperedges (one per atom). With k = 2, many of these hyperedges will have massive overlap (e.g., in a benzene ring, the 2-hop neighborhoods of all 6 atoms cover the entire ring). Would this lead to O(N²) redundant information? How could this redundancy be mitigated?

- In Eq. 7, the authors aggregate node structural encodings to the hyperedge level using mean aggregation. Have you tried other aggregation methods, such as sum, max, or attention-weighted aggregation?

- If the attention mechanism in Eqs. 8-9 are replaced with simple averaging. How much would the performance drop?

- MoleculeNet includes eight commonly used classification datasets. Why do you select only three of them? Also, could you include MoleculeNet regression datasets in your experiments?

**Questions:**

See weaknesses.

---

> ### Author Response · Authors · 2025-11-22
>
> **Response W1**: In the hypergraph construction state, there are indeed redundancy issues for overlapping hyperedges. We simply remove the totally overlapped hyperedge and leave one of them to fix the problem. We will add this detail in the latter version of this paper.
>
> **Response  W2**: We try the other aggregation methods on dataset BBBP and BACE, and list the results as follows:
>
> | Dataset            | **BBBP** |      |      | **BACE** |      |      |
> | ------------------ | -------- | ---- | ---- | -------- | ---- | ---- |
> | Aggregation Method | AUC      | Fid+ | Fid- | AUC      | Fid+ | Fid- |
> | Sum                | 96.8     | 17.6 | 48.0 | 86.3     | 17.7 | 43.9 |
> | Max                | 93.4     | 19.2 | 45.2 | 85.6     | 18.1 | 41.5 |
> | Attention Weighted | 97.0     | 18.4 | 48.3 | 88.3     | 16.2 | 43.8 |
> | Mean (Our Method)  | 97.2     | 17.1 | 48.2 | 90.4     | 15.3 | 44.2 |
>
> **Response W3**: We set up this ablation aspect and the results are listed in the following tables.
>
> | Dataset          | **BBBP** |      |      | **BACE** |      |      |
> | ---------------- | -------- | ---- | ---- | -------- | ---- | ---- |
> | Variant          | AUC      | Fid+ | Fid- | AUC      | Fid+ | Fid- |
> | Simple Averaging | 94.3     | 22.7 | 43.4 | 87.9     | 19.7 | 38.6 |
> | Our Method       | 97.2     | 17.1 | 48.2 | 90.4     | 15.3 | 44.2 |
>
> **Response W4** : Our current work focuses on explainability for binary molecular property prediction. Therefore, we select datasets from MoleculeNet that align with this task setting. Extending our framework to regression problems is an important direction, and we plan to investigate regression datasets in future work.

---

### Meta-Review · Area_Chair_FuEM · 2025-12-20

**Summary:**

The paper proposes HyperSEM, a multi-view hypergraph framework for self-explainable molecular property prediction, and the core idea of combining hyperedge-based atomic-group modeling with chemical-rule-guided multi-view learning is conceptually appealing. However, reviewers raised persistent concerns regarding the novelty of the hypergraph-based explanation paradigm, the clarity and completeness of methodological and experimental details, and the fairness and reproducibility of interpretability and efficiency comparisons, which were only partially addressed in the rebuttal. Given these unresolved issues, the submission would be reject.

**Reviewer Concerns:**

Reviewer nzLU raised technical questions regarding potential k-hop neighborhood redundancy, aggregation choices at the hyperedge level, the necessity of the attention mechanism, and the limited selection of MoleculeNet datasets. In the rebuttal, the authors clarified how redundancy across overlapping hyperedges is mitigated, added new datasets such as BBBP and BACE, and included additional ablation studies.

----

Reviewer AC1y engaged in a detailed exchange with the authors, including discussions on expanded baselines, clearer motivation, and experimental details. However, in the subsequent round of discussion, several new conceptual concerns remained unanswered. Specifically, the reviewer noted that the paper should provide an explicit and self-contained definition of “explanation,” and argued that the method may hold an inherent advantage given its motivation rather than offering a fundamentally novel contribution. They also questioned the claimed novelty of treating hyperedges as interpretable units and remained unconvinced about the practical usefulness of the proposed self-explainability, asking for concrete drawbacks of post-hoc explanation methods

---

Reviewer b3mD raised concerns regarding motivational sufficiency, limited novelty, the technical clarity of the proposed hypergraph-based design, and potential biases in explanation evaluation. Although the authors addressed several questions during the rebuttal, the reviewer’s follow-up response indicated that their main concerns remain unresolved. They argued that using hypergraph structures for explanation is already well explored and that applying it to property prediction does not constitute a substantial conceptual advance. They also questioned the unexpected behavior in the ablation study after removing the IB constraint and noted that fairness issues in explanation evaluation especially the mismatch between threshold-based and top-K explainers.

---

Reviewer fcrj raised a large number of detailed technical and methodological questions. Although the authors addressed some of these points in the initial rebuttal, the reviewer explicitly stated that many of their concerns remain unresolved, indicating that key issues in the paper’s formulation, implementation clarity, and experimental justification were not adequately addressed.

---

Reviewer LWZa raised concerns about notation clarity, baseline selection for interpretability, metric specification, implementation transparency, ablation interpretability, and the efficiency comparison with GradCAM. In the rebuttal, the authors addressed some of these points, clarified parts of the methodology, and added further explanations and experiments. However, the reviewer emphasized that code availability is still missing, that the procedure for converting soft attribution scores into hard explanatory subgraphs remains under-specified, and that the efficiency comparison is not entirely fair unless forward passes are treated consistently across methods.

**Reviewer Scores:**

Reviewer nzLU is likely to maintain a positive assessment following the rebuttal.

---

Reviewer AC1y’s core concerns about novelty and practical value remain unresolved.

---

Reviewer b3mD will maintain their original score.

---

Reviewer fcrj  is unlikely to change their original assessment

----

Reviewer LWZa explicitly indicated that resolving these remaining issues would justify a higher score, but since they are not yet fully addressed,  the score is not increased at this stage.

---

### Decision · Program_Chairs · 2026-01-26

Reject